# Pay Less Attention to Function Words for Free Robustness of Vision-Language Models

**Qiwei Tian, Chenhao Lin,**[*] **Zhengyu Zhao, Chao Shen**
School of Cyber Science and Engineering, Xi'an Jiaotong University, Xi'an 710049, China
`michaeltqw@stu.xjtu.edu.cn`, {`linchenhao, zhengyu.zhao`}`@xjtu.edu.cn`,
`chaoshen@mail.xjtu.edu.cn`

## Abstract

To address the trade-off between robustness and performance for robust VLM, we observe that function words could incur vulnerability of VLMs against cross-modal adversarial attacks, and propose Function-word De-Attention (FDA) accordingly to mitigate the vulnerability brought by function words. Inspired by differential transformers, our FDA calculates the original and the function-word cross-attention within attention heads, and differentially subtracts the latter from the former for more robust alignment. Comprehensive experiments include 2 SOTA baselines under 6 different attacks on 2 downstream tasks, 3 datasets, and 3 models. Overall, our FDA yields an average 18/13/53% ASR drop with only 0.2/0.3/0.6% performance drops on the 3 tested models on retrieval, and a 90% ASR drop with a 0.3% performance gain on visual grounding. We demonstrate the scalability, generalization, and zero-shot performance of FDA experimentally, as well as in-depth ablation studies and analysis. Code is available at `https://github.com/michaeltian108/FDA`.

## 1 Introduction

Building robust vision-language models (VLMs) has garnered profound academic focus because of the necessity of defending VLMs against various adversarial attacks. To this end, many works have been proposed to defend models against adversarial images by enhancing model robustness (Schlarmann et al., 2024; Mao et al., 2022), purifying perturbations (Yoon et al., 2021; Lei et al., 2025), or detecting potential adversaries (Metzen et al., 2017; Klingner et al., 2022). Among them, adversarial training (AT) shows superior enhancement in the robustness of VLMs, at cost of significant performance drops and computational costs.

In this paper, we propose to enhance adversarial robustness from an overlooked yet intuitive perspective. We hypothesize that, due to the innate semantic gaps between the clean image and targeted texts, cross-modal perturbation are likely guided towards closer tokens (i.e., smaller semantic gaps). In this regard, function words, which carry little semantic information but have high word frequency, become *more approachable than other tokens, incurring unwanted vulnerability*. To demonstrate the above hypothesis, we record and visualize the relative clean R@1 and Attack Success Rate (ASR) when removing verbs, adjectives, and function words, compared to not removing. As shown in the left of Fig.1, only function words *reduce* ASR among all word types and cause the least clean performance drop. The results qualitatively confirms the impact of function words regarding in bringing adversarial vulnerability. We further provide the attention maps (Selvaraju et al., 2017) to visualize the impact of removing different words, presented in the right of Fig.1. All above results validate our above hypothesis, implying that **function words carry certain adversarial semantic**, and **removing them properly could tighten the cross-modal alignment, defending VLMs against attacks**.

Consequently, inspired by the setting of differential transformers (Ye et al., 2024) and differential amplifiers, we propose Function-word De-Attention (FDA) as the first method to build robust VLMs by refining vision-language alignment. Specifically, our FDA works by deploying a parallel pipeline on multi-attention heads within fusion-encoders, calculating the cross-attention between function

---

[*]Corresponding author.

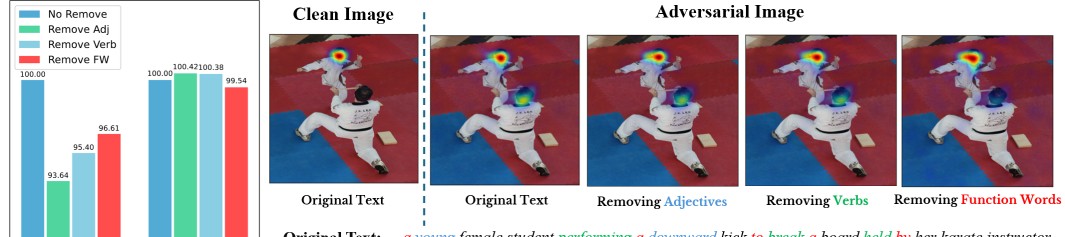

Figure 1: **Left:** Histograms on of relative clean R@1 and ASR of *No remove*, *Removing Adj*, *Verbs* and *Function words*. Results are collected from Flickr30k retrieval testset on ALBEF. **Results qualitatively show that removing function words helps reduce ASR and maintain clean performance**. **Right:** Attention maps of VLM under white-box attacks through perturbed images. The texts are given at the bottom, with words marked by corresponding colors. VLM correctly recognizes the corresponding character on the clean image given the token `her`, but gets distracted towards the incorrect character on the adversarial image. Only removing function words helps the VLM 'looks back at' the correct character.

words and the input images, i.e., distractions. We further softmax along the dimensions of visual and textual tokens to highlight the most misleading textual/visual tokens. Finally, we subtract the above distractions from the original attention for the output. To validate the effectiveness of FDA, we conduct comprehensive experiments on two SOTA baselines, 3 models, 2 tasks, 3 datasets, and 6 attacks. Overall, our FDA yields an average 18/13/53% ASR drop with only 0.2/0.3/0.6% performance drops on the 3 tested models for retrieval, and a 90% ASR drop with *better* clean performance on grounding. We also find that FDA improves the zero-shot performance.

Overall, our contributions are summarized as follows:

1) We identify that function words are vulnerability for VLMs and subsequently propose Function-word De-Attention (FDA) to pay less attention to function words for more aligned vision-language models with free robustness.

2) We conduct comprehensive experiments on two SOTA baselines, 3 models, 2 tasks, and 3 datasets, under 6 attacks, and validate the effectiveness of FDA in enhancing robustness while preserving performance.

3) We provide ablation studies to show the hyperparameters insensitivity, backbone generalization, and zero-shot performance enhancement of our method with in-depth analysis.

## 2 RELATED WORK

**Adversarial attacks on vision-language models.** In light of the advancement in VLMs (Li et al., 2021; 2022; Radford et al., 2021; Bai et al., 2023; Team et al., 2024; Bai et al., 2025; Yang et al., 2025; Liu et al., 2023),several aspects (Li et al., 2025b;a; Zha et al., 2025), such as token compression and broader capabilities, have garnered enormous research interests. Among them, adversarial attacks on VLMs become the most threatening direction as they fool VLMs into incorrect or misleading outputs. Recent studies on white-box attacks (Croce & Hein, 2020) have exhibited impressive results. Besides, black-box attacksZhang et al. (2022); Lu et al. (2023); Yin et al. (2023); He et al. (2023); Tian et al. (2026) have also demonstrated significant effectiveness against pre-trained VLMs through transferable cross-modal attacks.

**Adversarial Defense on vision-language models** For defenses, adversarial training (AT) (Rice et al., 2020; Zhang et al., 2019; Tian et al., 2024) has exhibited significant effectiveness in defending models against various adversarial attacks against classification, retrieval, etc. Several AT-based methods (Schlarmann et al., 2024; Mao et al., 2022) have demonstrated impressive robustness boost on CLIP models. However, AT is notoriously well-known for downgrading performance significantly due to the inclusion of adversarial examples into training. Although Schlarmann et al. (2024) proposed FARE to use the visual embeddings of vanilla models as supervision to balance the

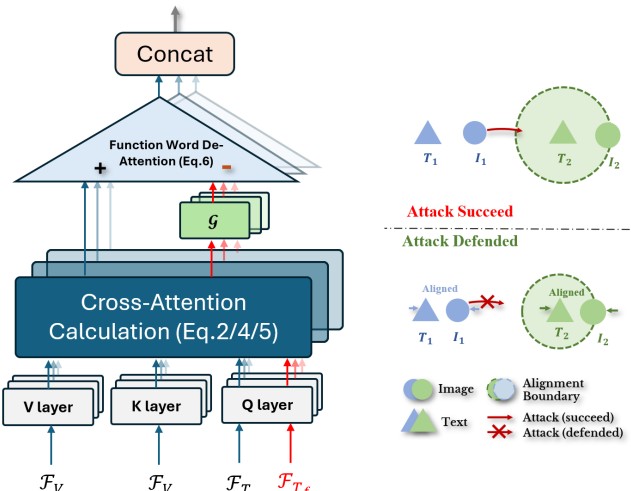

Figure 2: **Left:** An illustration of our Function-word De-Attention (FDA) method. On the existing process of attention calculation, which uses $\mathcal{F}_V$ and $\mathcal{F}_T$, we add a parallel pipeline to calculate the attentions between function words $\mathcal{F}_{T_f}$ and the images $\mathcal{F}_V$. Afterwards, the function-attention passes a control gate $\mathcal{G}$ before entering the FDA module (triangle) differentially to subtract distractions as presented in Eq.6. **Right:** An illustration on how FDA defends attacks. We speculate that attacks can cross the boundary easily to cause misalignments in vanilla models (top), and by removing function-word distractions, models can learn a tightened embedding (bottom), preventing such misalignments.

trade-off between clean performance and robustness, the performance drops remain considerably noticeable. Besides, the high computational costs also hinder broader applications in practice.

## 3 METHODOLOGY

In this section, we first provide a brief preliminary for the original calculation pipeline of cross-attention and introduce our Function-words De-Attention (FDA).

### 3.1 PRELIMINARY

For a given textual/visual encoder $\mathcal{T}/\mathcal{V}$, input images $I$ and texts $T$ are fed into respective encoders with their attention masks $\mathcal{M}_T/\mathcal{M}_I$ for the embeddings $\mathcal{F}_T, \mathcal{F}_V \in \mathbb{R}^{d_k}$:

$$\mathcal{F}_T = \mathcal{T}(T, \mathcal{M}_T), \quad \mathcal{F}_V = \mathcal{V}(I, \mathcal{M}_I) \tag{1}$$

Then, cross-attention scores are calculated by inputting these hidden states into the fusion encoder:

$$Att^{L,H} = softmax\Big(\frac{Q(\mathcal{F}_T)K(\mathcal{F}_V)}{\sqrt{d_k}}, dim = -1\Big) \cdot V(\mathcal{F}_V) \tag{2}$$

where $Q/K/V$ is the query/key/value layers, and $L, H$ is the index of layers and attention heads.

### 3.2 FUNCTION-WORD DE-ATTENTION (FDA)

Built upon our previous hypothesis that function words are potential distractions in vision-language alignment, we propose Function-word De-Attention (FDA) to remove such distractions: we add a parallel pipeline upon the existing cross-attention calculation to specifically acquire the cross-attention between function words and the input images, namely the distraction, and then subtract them from the original attention. An illustration of our FDA is given in Fig.2. We first parallelly extract the features of all function words (denoted as $T_f$) within the input texts by masking all other tokens, excluding `[CLS]`:

$$\mathcal{F}_{T_f} = \mathcal{T}(T, \mathcal{M}_{T_f}), \forall T_f \in \mathcal{D} \tag{3}$$

where $M_{t_f}$ is the function word mask, and $\mathcal{D}$ is the function word dictionary, which is a dictionary shortlisted from the stopwords list in (Li et al., 2020). We then use a parallel pipeline to calculate function words' attention scores:

$$S_{T_f}^{L,H} = \frac{Q(\mathcal{F}_{T_f})K(\mathcal{F}_V)^T}{\sqrt{d_k}} \tag{4}$$

With the function words attention scores, we then conduct softmax along the dimensions of visual tokens and textual tokens, respectively. In this way, we highlight the visual tokens with false activation under token words or the most misleading tokens with the largest visual activation:

$$\tilde{Att}_t^{L,H} = softmax\big(S_{T_f}^{L,H}, dim = -1\big)V, \qquad \tilde{Att}_v^{L,H} = softmax\big(S_{T_f}^{L,H}, dim = -2\big)V \tag{5}$$

Afterwards, we subtract both attention from $Att$ individually and take the minimum value as the final attention scores, with $\mathcal{G}$ being a control gate for subtraction intensity.

$$\hat{Att}^{L,H} = \min\big(Att^{L,H} - \mathcal{G}(\tilde{Att}_t^{L,H}), Att^{L,H} - \mathcal{G}(\tilde{Att}_v^{L,H})\big) \tag{6}$$

Finally, after complete calculation of FDA, denoted as $\mathbb{D}(\cdot)$, we concatenate the attention scores from all attention heads:

$$\hat{\mathbf{Att}}^{L,H} = Concat\Big(\mathbb{D}\Big(Q(\mathcal{F}_T, \mathcal{F}_{T_f}), K(\mathcal{F}_V), V(\mathcal{F}_V)\Big); ...\Big) \tag{7}$$

FDA can be flexibly implemented on any number of fusion layers/attention heads, as each may specialize differently (Kang et al., 2025). A general intuition is to remove these distractions in the early layers instead of the later ones to avoid possible 'absorption', but not so exquisitely upfront that it may undermine the contextual integrity of the textual inputs. We provide ablation and analysis in Sec.4.3.

## 4 EXPERIMENTS

**Vision-Language Tasks&Datasets.** To thoroughly evaluate the performance and robustness of FDA, we incorporate several downstream tasks, including Text-to-Image/Image-to-Text Retrieval (T2IR/I2TR) and Visual Grounding (VG). For datasets, we use the Flickr30k(Plummer et al., 2015) and MSCOCO (Lin et al., 2014) dataset for retrieval, and RefCOCO+ (Yu et al., 2016) for VG.

**Models.** For T2IR/I2TR, we test our method on the ALBEF (Li et al., 2021), TCL (Yang et al., 2022), and BLIP(Li et al., 2022), using 14M/14M/124M pretrained images, respectively.For VG, we use ALBEF . All models use the ViT-B/32 (Dosovitskiy et al., 2021) as visual encoders and BERT (Devlin et al., 2019) as textual encoders. Specifically, TCL shares the same backbones as ALBEF but uses a different training strategy (triplet contrastive learning), while BLIP uses a larger pre-trained encoder with an extra decoder. CLIP (Radford et al., 2021) is not included due to the absence of fusion encoders.

**Baselines.** As for baselines, we adopt the two SOTA methods for robust CLIP, i.e., TeCoA (Mao et al., 2022) and FARE (Schlarmann et al., 2024), on all the models as adversarial fine-tuning baselines. To account for the robustness and accuracy trade-off, we lower the perturbation strength of each method to ensure similar clean performance as our FDA, such that TeCoA and FARE serve as a reference to compare robustness. Specifically, we use $\epsilon = 1$ for TeCoA and FARE. For Text-to-Image/Image-to-Text Retrieval, we adversarially fine-tune using TeCoA and FARE for 4/1/1 epochs for ALBEF/TCL/BLIP. For Visual Grounding, we adversarially train models with both TeCoA and FARE for 1 epoch, as both methods incur significant performance drops.

**Attacks.** To thoroughly evaluate the robustness of our models, we test all models with three adversarial attacks and use the average of all attacks for robustness evaluations. Specifically, we use

Table 1: Attack success rate (ASR) of PGD/APGD/MAPGD (masked APGD) against for *Text-to-Image/Image-to-Text Retrieval* (T2IR/I2TR) on Flickr30k and COCO. Results are presented in percentage (%). ↑ /↓ indicates increased/decreased $\Delta_{ASR}$ (higher values preferred). † indicates higher performance than clean models. (Full results are given in Sec.C of the Appendix.) **Our FDA consistently shows the best performance and overall robustness on ALL models.**

| Dataset | VLM | $l_\infty$ | Defense | Text-to-Image Retrieval | | | | Image-to-Text Retrieval | | | | ASR drop |
|---|---|---|---|---|---|---|---|---|---|---|---|---|
| | | | | Clean (R@1) | ASR (↓) | | | Clean (R@1) | ASR (↓) | | | |
| | | | | | PGD | APGD | MAPGD | | PGD | APGD | MAPGD | $\Delta_{ASR}$ ↑ |
| Flickr | ALBEF | 2/255 | No Defense | 95.90 | 3.38 | 14.68 | 65.88 | 85.60 | 0.71 | 14.98 | 58.85 | - |
| | | | TeCoA | 91.20 | 2.56 | 19.39 | 73.12 | 81.44 | 0.55 | 17.45 | 61.30 | ↓ 3.02 |
| | | | FARE | 91.10 | **2.46** | 17.29 | 70.15 | 81.48 | 0.55 | 16.55 | 65.90 | ↑ 5.36 |
| | | | FDA | **95.60** | 3.37 | **12.44** | **58.66** | **85.50** | **0.35** | **12.55** | **51.35** | ↑**22.26** |
| | | 4/255 | No Defense | 95.90 | 8.72 | 16.09 | 80.92 | 85.60 | 7.20 | 15.89 | 77.14 | - |
| | | | TeCoA | 91.20 | 9.13 | 19.34 | 85.48 | 81.44 | 4.60 | 18.45 | 79.60 | ↓ 2.15 |
| | | | FARE | 91.10 | 9.27 | 18.60 | 86.25 | 81.48 | 5.20 | 18.35 | 80.60 | ↓ 2.48 |
| | | | FDA | **95.60** | **7.90** | **13.70** | **75.80** | **85.50** | 4.90 | **13.70** | **71.00** | ↑**14.69** |
| | TCL | 2/255 | No Defense | 94.90 | 10.29 | 70.55 | 66.66 | 84.02 | 4.11 | 65.58 | 60.79 | - |
| | | | TeCoA | 92.10 | 11.08 | 66.31 | **59.11** | 80.40 | 4.10 | 70.80 | **46.85** | ↑ 2.78 |
| | | | FARE | 91.70 | 11.72 | 67.47 | 60.98 | 78.22 | 4.60 | 61.25 | 47.85 | ↑ 1.09 |
| | | | FDA | **94.40** | **8.52** | **48.38** | 68.48 | **83.82** | **3.30** | **44.50** | 57.50 | ↑**14.29** |
| | | 4/255 | No Defense | 94.90 | 37.66 | 81.11 | 81.63 | 84.02 | 29.72 | 78.36 | 73.10 | - |
| | | | TeCoA | 92.10 | 44.29 | 80.62 | 80.08 | 80.40 | 35.40 | 76.60 | 67.95 | ↓ 4.16 |
| | | | FARE | 91.70 | 46.21 | 81.03 | **79.64** | 78.22 | 38.05 | 76.95 | **67.35** | ↓ 6.42 |
| | | | FDA | **94.40** | **30.36** | **58.64** | 86.24 | **83.82** | **24.25** | **56.50** | 77.80 | ↑**13.55** |
| | BLIP | 2/255 | No Defense | 97.20 | 25.10 | 63.26 | 50.19 | 87.30 | 11.83 | 60.08 | 44.35 | - |
| | | | TeCoA | 90.30 | 19.28 | 59.38 | 48.67 | 78.04 | 8.85 | 47.80 | 37.15 | ↑15.70 |
| | | | FARE | 89.70 | 20.24 | 66.53 | 54.92 | 77.72 | 10.00 | 58.00 | 46.65 | ↑ 3.09 |
| | | | FDA | **96.50** | **7.66** | **18.96** | **40.98** | **86.84** | **5.50** | **13.75** | **35.00** | ↑**51.60** |
| | | 4/255 | No Defense | 97.20 | 61.18 | 86.39 | 71.27 | 87.30 | 67.00 | 86.08 | 71.60 | - |
| | | | TeCoA | 90.30 | 62.39 | 88.85 | 75.69 | 78.04 | 62.35 | 87.35 | 72.30 | ↓ 1.09 |
| | | | FARE | 89.70 | 66.29 | 92.35 | 80.49 | 77.72 | 67.30 | 90.50 | 82.50 | ↓ 7.04 |
| | | | FDA | **96.50** | **15.86** | **28.37** | **60.64** | **86.84** | **14.45** | **16.30** | **55.50** | ↑**56.36** |
| COCO | ALBEF | 2/255 | No Defense | 77.60 | 0.95 | 11.01 | 30.47 | 60.70 | 0.35 | 8.86 | 19.40 | - |
| | | | TeCoA | 68.04 | 0.72 | 18.56 | 34.23 | 53.07 | 0.15 | 13.05 | 18.89 | ↑ 2.87 |
| | | | FARE | 69.28 | **0.26** | 22.68 | 32.71 | 53.58 | **0.02** | 14.59 | **16.76** | ↑ 0.53 |
| | | | FDA | **77.70**† | 0.84 | **9.65** | **27.60** | **60.63** | 0.28 | **8.03** | 18.02 | ↑ **9.28** |
| | | 4/255 | No Defense | 77.60 | 4.71 | 14.48 | 51.20 | 60.70 | 2.41 | 12.18 | 36.17 | - |
| | | | TeCoA | 68.04 | 1.57 | 25.69 | 59.90 | 53.07 | 0.36 | 20.57 | 40.37 | ↓ 3.25 |
| | | | FARE | 69.28 | **1.40** | 32.34 | 63.45 | 53.58 | **0.35** | 24.35 | 39.61 | ↓16.08 |
| | | | FDA | **77.70**† | 3.82 | **11.87** | **44.92** | **60.63** | 2.05 | **10.57** | **32.83** | ↑**14.43** |

Projected Gradient Descent (Madry et al., 2017) and AutoAttack (Croce & Hein, 2020), denoted as PGD and APGD. As for the adaptive attack, we apply function word masks to the input texts for APGD to evade our FDA, denoted Masked APGD (MAPGD). **All attacks are fully white-box**, i.e., the attackers can access FDA. For each attack, we follow the settings in Mao et al. (2022) and Schlarmann et al. (2024) to attack images with perturbation bounded by $l_\infty = \frac{2}{255}, \frac{4}{255}$. Specifically, for targeted attacks in T2IR/I2TR, we apply a circular shift targeted to ensure non-overlapping unmatched targets. For targeted attacks in VG, we follow the settings of Gao et al. (2024) to not apply patched attacks.

**Metrics.** We use the common metric, Attack Success Rate (ASR), to indicate the efficacy of all adversarial attacks. For an ASR on a defended model and the baseline model, denoted as $ASR_M$ and $ASR_B$, respectively, we calculate the relative ASR change in percentage using $\Delta_{ASR} = \frac{ASR_B - ASR_M}{ASR_B} \times 100\%$. Consequently, a positive/larger $\Delta_{ASR}$ indicates improved/stronger robustness, while a negative/lower $\Delta_{ASR}$ implies decreased/weaker robustness, with 0% (100%) meaning no robustness gain (completely defended). Details are given in the Sec.B of the Appendix.

**Implementation Details.** Since our FDA parallelly computes distracted attention for subtraction, fine-tuning models with FDA is identical to downstream fine-tuning without extra modifications or parameters. Following the settings in Li et al. (2021), we fine-tune the model by 10 epochs and use the last-epoch model for all tasks/models. For the layer index, we use $L^{0-1}$ and $H^{0-5}$ for all models/tasks/datasets, with ablation studies on the selection of the layer and attention head indices for all models and tasks in Sec.4.3.

Table 2: Attack success rate (ASR) of PGD/APGD/MAPGD against for **Visual Grounding** (VG) on RefCOCO+. Results are presented in percentage (%). ↑/↓ indicates increased/decreased $\Delta_{ASR}$ (higher values preferred). † indicates higher performance than clean models. (Full results are given in Sec.C of the Appendix.) **Our FDA consistently leads in performance and overall robustness on ALL models.**

| $l_\infty$ | Defense | Clean (Acc) | | | ASR on Test A Split (↓) | | | ASR on Test B Split (↓) | | | Avg ASR drop |
|---|---|---|---|---|---|---|---|---|---|---|---|
| | | Val_d | Test A | Test B | PGD | APGD | MAPGD | PGD | APGD | MAPGD | $\Delta_{ASR}$ ↑ |
| 2/255 | No Defense | 58.50 | 65.90 | 46.30 | 6.70 | 11.16 | 11.16 | 6.07 | 7.08 | 7.42 | - |
| | TeCoA | 57.20 | 64.70 | 45.00 | 6.81 | 7.72 | 8.01 | 3.39 | 6.28 | 6.10 | ↑21.21 |
| | FARE | 56.40 | 64.20 | 44.70 | 6.13 | 10.42 | 9.96 | 4.54 | 6.72 | 6.21 | ↑12.08 |
| | FDA | **58.10** | **66.80†** | **46.10** | **1.36** | **2.41** | **1.80** | **0.00** | **0.00** | **0.00** | ↑**93.16** |
| 4/255 | No Defense | 58.50 | 65.90 | 46.30 | 7.89 | 11.16 | 11.75 | 4.39 | 8.06 | 8.06 | - |
| | TeCoA | 57.20 | 64.70 | 45.00 | 6.57 | 8.17 | 8.46 | 3.56 | 6.10 | 6.44 | ↑21.63 |
| | FARE | 56.40 | 64.20 | 44.70 | 6.74 | 9.66 | 10.27 | 4.03 | 7.06 | 6.55 | ↑13.28 |
| | FDA | **58.10** | **66.80†** | **46.10** | **1.50** | **2.10** | **2.10** | **0.34** | **0.00** | **0.00** | ↑**91.50** |

## 4.1 MAIN RESULTS

In this section, we compare the robustness of FDA and other baselines on T2IR/I2TR and VG tasks.

**T2IR/I2TR.** Results of T2IR/I2TR on ALBEF, TCL, and BLIP for all methods are given in Table.1. Overall on Flickr, our FDA consistently exhibits the **best robustness** with the **best clean performance** on **all models** over **all other baselines** for $\epsilon = \frac{2}{255}/\frac{4}{255}$, yielding a 22.26/14.69%, 14.29/13.55%, and 51.60/56.36% average ASR drop over all 3 attacks on ALBEF/TCL/BLIP, with a negligible 0.30/0.10%, 0.50/0.22%, and 0.70/0.46% performance drops in R@1 for T2IR/I2TR on each model, respectively. On MSCOCO, similar patterns exist as our FDA boosts the ASR drop by 9/14% for $\epsilon = \frac{2}{255}/\frac{4}{255}$ with a 0.1% clean performance boost on T2IR and a 0.07% drop on I2TR.

*i.* Attack-wise, on Flickr, our FDA exhibits the best defense against PGD and APGD in 22 out of 24 results, leading TeCoA/FARE by 60/65% on the BLIP model, demonstrating the effectiveness of FDA in enhancing robustness against various attacks. For the strongest adaptive attack, MAPGD, our FDA maintains its lead over TeCoA and FARE on ABLEF and BLIP, with an average lead by over 10%. Although our FDA shows more vulnerability against APGD on the TCL model, it retains the best comprehensive robustness of the other two baselines, yielding a 10-20% lead for $\epsilon = 2/4$. TeCoA/FARE become ineffective for attacks with $\epsilon = 4$, while our FDA retains its effectiveness facing stronger attacks. Similar trends also exist on MSCOCO.

*ii.* Performance-wise, all baseline methods (i.e., TeCoA and FARE) suffered from a performance drop by an average 4/3/9% on ALBEF/TCL/BLIP. Nevertheless, our FDA only causes minor or little drops of less than 1% for all models, yielding a lead of TeCoA and FARE by approximately 4%, 3%, and 7% on average, demonstrating the feasibility of paying less attention to function words for free robustness.

*iii.* Scalability-wise, we find that the effectiveness of FDA benefits significantly as the model scales: on ALBEF/TCL, which uses 14M pre-trained images, FDA enhances robustness of each model by roughly 15%; while on BLIP, which uses 124M pre-trained images, FDA achieves an impressive 54% overall increase in $\Delta_{ASR}$. We attribute the enhancement to the capability of the backbone model, which enables the encoders to better capture visual clues.

**Grounding.** Similar patterns persist for VG as shown in Table.2. Our FDA achieves almost complete defense for all attacks, yielding an over 90% ASR drop while **performing better on clean examples** than the vanilla model. Specifically, FDA shows 93.16/91.50% ASR drop for $\epsilon = 2/4$. While TeCoA and FARE show comparative clean performance, they achieve 21.21/21.63% and 12.08/13.28% ASR drop, respectively, with an over 1% drop on clean examples. These results confirm the efficacy of FDA in enhancing robustness for similar/better clean performance.

## 4.2 UNTARGETED ATTACKS

We further evaluate the robustness against untargeted attacks. Thus, we retrained all models using TeCoA/FARE and their combination with our FDA to validate the effectiveness of FDA in defending against untargeted attacks.

Table 3: Robustness evaluations on ALBEF using FDA as a plug-and-play tool with TeCoA and FARE against untargeted attacks for Text-to-Image/Image-to-Text Retrieval. Results are averaged over T2IR and I2TR. Full results are provided in Sec.D of the Appendix. **FDA consistently boosts clean performance and/or robustness against all attacks.**

| VLM | Defense | Clean (R@1) | | Average ASR $2/255$ (↓) | | | Average ASR $4/255$ (↓) | | | Avg ASR drop |
|---|---|---|---|---|---|---|---|---|---|---|
| | | T2IR | I2TR | PGD | APGD | MAPGD | PGD | APGD | MAPGD | $\Delta_{ASR}$ ↑ |
| ALBEF | No Defense | 95.90 | 85.60 | 72.67 | 68.13 | 63.19 | 94.71 | 83.81 | 81.75 | - |
| | TeCoA | 92.30 | 81.40 | 75.84 | 64.09 | 61.65 | **97.49** | 81.41 | 82.69 | ↑ 0.47 |
| | TeCoA + FDA | **92.50** | **81.86** | **75.52** | **63.22** | **60.44** | 97.57 | **80.84** | **82.01** | ↑ **1.31** |
| | FARE | 91.20 | 80.76 | **69.87** | 48.18 | **44.00** | 96.43 | 75.79 | 75.79 | ↑13.09 |
| | FARE + FDA | **91.40** | **80.80** | 70.70 | **47.95** | 44.54 | **96.42** | **74.84** | **73.57** | ↑**13.46** |
| BLIP | No Defense | 97.20 | 87.30 | 78.17 | 77.08 | 67.65 | 99.80 | 94.01 | 89.82 | - |
| | TeCoA | **81.50** | **68.00** | 48.01 | 41.23 | 38.16 | 95.37 | 75.41 | 72.39 | ↑28.23 |
| | TeCoA + FDA | 80.40 | 67.78 | **43.80** | **38.20** | **35.63** | **94.26** | **72.20** | **69.67** | ↑**32.18** |
| | FARE | 89.70 | 77.72 | 47.51 | 53.62 | 51.45 | 90.37 | 78.71 | 77.01 | ↑22.36 |
| | FARE + FDA | **89.80** | 77.72 | **45.07** | **49.54** | **46.97** | **89.96** | **76.11** | **74.15** | ↑**25.91** |

Table 4: Robustness evaluations of FDA as a plug-and-play tool with TeCoA and FARE against untargeted attacks for Visual Grounding. Results are averaged over Test-A and Test-B. Full results are provided in Sec.D of the Appendix. **FDA consistently boosts clean performance and robustness against all attacks.**

| Defense | Clean (Acc) | | | Average ASR $2/255$ (↓) | | | Average ASR $4/255$ (↓) | | | Avg ASR drop |
|---|---|---|---|---|---|---|---|---|---|---|
| | Val_d | Test A | Test B | PGD | APGD | MAPGD | PGD | APGD | MAPGD | $\Delta_{ASR}$ ↑ |
| No Defense | 58.50 | 65.90 | 46.30 | 27.46 | 20.06 | 19.82 | 32.33 | 23.48 | 23.31 | - |
| TeCoA | 57.20 | 64.70 | 45.00 | **9.67** | **12.68** | 12.80 | **9.64** | 16.22 | 16.43 | ↑39.68 |
| TeCoA + FDA | 57.00 | **64.90** | **45.30** | 10.37 | 12.85 | **12.01** | 9.84 | **15.22** | **15.67** | ↑**40.30** |
| FARE | 56.40 | **64.20** | 44.70 | **10.56** | 14.45 | 14.49 | 10.79 | 16.70 | 16.97 | ↑34.69 |
| FARE + FDA | 56.10 | 63.70 | 44.70 | 11.25 | **13.02** | **13.05** | **10.46** | **15.20** | **15.50** | ↑**39.35** |

Table 5: Ablation studies on the encoders and dictionary of FDA. We use T2IR/I2TR for evaluation.

| Defense | Clean (R@1) | | | Average ASR $2/255$ (↓) | | | Average ASR $4/255$ (↓) | | | Avg ASR drop |
|---|---|---|---|---|---|---|---|---|---|---|
| | T2IR | I2TR | Avg | PGD | APGD | MAPGD | PGD | APGD | MAPGD | $\Delta_{ASR}$ ↑ |
| w/o FDA | 95.90 | 85.60 | 90.75 | 2.04 | 14.83 | 62.37 | 7.96 | 15.99 | 79.03 | - |
| $\mathcal{T}$ | 95.10 | 85.28 | 90.19 | 2.10 | 21.86 | 8.15 | 10.66 | 24.70 | 17.30 | ↓ 2.54 |
| $\mathcal{T}$ & $\mathcal{H}$ | 93.80 | 85.00 | 89.40 | 2.01 | 17.61 | 15.82 | 9.06 | 21.91 | 20.99 | ↑15.61 |
| $\mathcal{H}$ | **95.60** | **85.50** | **90.55** | 1.86 | 12.50 | 55.00 | 6.40 | 13.70 | 73.12 | ↑**18.48** |
| Full Dict | 95.10 | 84.46 | 89.78 | 2.03 | 13.54 | 56.60 | 6.46 | 14.47 | 74.65 | ↑ 4.22 |
| Shortlisted Dict | **95.40** | **85.40** | **90.40** | **1.71** | 13.60 | 56.78 | 6.92 | **14.15** | 75.07 | ↑ **6.45** |

For T2IR/I2TR, as presented in Table.3. Overall, FDA consistently boosts the robustness of TeCoA and FARE for all untargeted attacks on all models. Specifically, the scalability of FDA also persists with TeCoA/FARE: both methods benefit more from FDA on the larger backbone of BLIP, yielding a 4/3% robustness boost. Furthermore, we notice that FDA also boosts the clean performance of both methods on ALBEF considerably. For VG, we observe identical patterns: implementing FDA yields a solid robustness gain. For example, FARE experiences a significant robustness enhancement regarding untargeted attacks by 5%. In sum, our FDA compatibility works with both TeCoA and FARE to further **boost their robustness against untargeted attacks**.

## 4.3 ABLATION STUDY

All results are conducted on targeted attacks, hyperparameters of FDA: encoder, dictionary, layer/head, variant of FDA, and zero-shot performance. (See full results in Sec.E of the Appendix.)

**Hyperparameters.** The implementation of FDA, especially the macro-hyperparameters influencing where to implement, would largely impact the subsequent performance of models. We first provide relative ablation studies to help understand the mechanics and design of our FDA.

Table 6: Ablation studies on the layer/head index $L/H$ of FDA on Text-to-Image/Image-to-Text Retrieval on ALBEF, TCL and BLIP. Results are averaged over T2IR/I2TR. **Shallower layers/heads (smaller $L/H$) consistently outperform over others on retrieval tasks.**

| VLM | Defense | Clean (R@1) | | | Average ASR $2/255$ ($\downarrow$) | | | Average ASR $4/255$ ($\downarrow$) | | | Avg ASR drop |
|---|---|---|---|---|---|---|---|---|---|---|---|
| | | T2IR | I2TR | Avg | PGD | APGD | MAPGD | PGD | APGD | MAPGD | $\Delta_{ASR}$ $\uparrow$ |
| **ALBEF** | w/o FDA | 95.90 | 85.60 | 90.75 | 2.04 | 14.83 | 62.37 | 7.96 | 15.99 | 79.03 | - |
| | $L^{all}, H^{all}$ | 95.50 | 85.54 | 90.52 | 2.06 | 14.82 | 60.98 | 7.82 | 16.15 | 80.70 | ↑ 1.56 |
| | $L^{all}, H^{6-11}$ | 95.00 | 84.96 | 89.98 | 2.31 | 17.76 | 65.95 | 7.91 | 19.46 | 83.70 | ↓ 8.70 |
| | $L^{all}, H^{0-5}$ | 95.40 | 85.40 | 90.40 | 1.71 | 13.60 | 56.78 | 6.92 | 14.15 | 75.07 | ↑ 6.45 |
| | $L^{0}, H^{0-5}$ | 95.60 | 85.50 | **90.55** | 1.86 | 12.50 | 55.00 | 6.40 | 13.70 | 73.12 | ↑**18.48** |
| | $L^{0-1}, H^{0-5}$ | 95.40 | 85.32 | 90.36 | 1.81 | 12.30 | 54.87 | 6.17 | 13.45 | 72.71 | ↑16.91 |
| **TCL** | w/o FDA | 94.90 | 84.02 | 89.64 | 7.20 | 68.07 | 62.37 | 33.69 | 79.74 | 79.03 | - |
| | $L^{all}, H^{0-5}$ | 94.10 | 83.98 | 89.04 | 6.17 | 54.34 | 65.59 | 29.24 | 66.24 | 84.47 | ↑ 8.52 |
| | $L^{0}, H^{0-5}$ | 94.40 | 83.82 | **89.11** | 6.06 | 46.44 | 62.99 | 27.31 | 57.57 | 82.02 | ↑**13.92** |
| | $L^{0-1}, H^{0-5}$ | 94.20 | 83.96 | 89.08 | 6.42 | 48.64 | 64.82 | 28.22 | 61.30 | 83.44 | ↑11.14 |
| **BLIP** | w/o FDA | 97.20 | 87.30 | 92.25 | 18.46 | 61.67 | 47.27 | 64.09 | 86.23 | 71.44 | - |
| | $L^{all}, H^{0-5}$ | 96.50 | 86.94 | 91.72 | 16.60 | 22.74 | 43.06 | 61.09 | 31.38 | 65.19 | ↑26.46 |
| | $L^{0}, H^{0-5}$ | 96.80 | 86.86 | **91.83** | 6.43 | 17.41 | 39.79 | 15.85 | 23.51 | 59.32 | ↑52.51 |
| | $L^{0-1}, H^{0-5}$ | 96.70 | 86.84 | 91.77 | 6.58 | 16.36 | 37.99 | 15.15 | 22.34 | 58.07 | ↑**53.98** |

Table 7: Ablation studies on the layer/head index $L/H$ of FDA on VG on ALBEF, TCL, and BLIP. Results are given in average.

| Defense | Clean (Acc) | | | | Average ASR $2/255$ ($\downarrow$) | | | Average ASR $4/255$ ($\downarrow$) | | | Avg ASR drop |
|---|---|---|---|---|---|---|---|---|---|---|---|
| | Val_d | Test A | Test B | Avg | PGD | APGD | MAPGD | PGD | APGD | MAPGD | $\Delta_{ASR}$ $\uparrow$ |
| w/o FDA | 58.50 | 65.90 | 46.30 | 56.90 | 6.38 | 9.12 | 9.29 | 6.14 | 9.61 | 9.90 | - |
| $L^{all}, H^{0-5}$ | 57.90 | 65.80 | 46.40 | 56.70 | 1.02 | 2.06 | 2.27 | 0.77 | 2.38 | 2.22 | ↑81.83 |
| $L^{0}, H^{0-5}$ | 58.00 | 65.90 | 46.40 | 56.77 | 1.72 | 3.09 | 2.93 | 1.85 | 2.60 | 2.76 | ↑71.43 |
| $L^{0-1}, H^{0-5}$ | 58.10 | 66.80 | 46.10 | **57.00** | 0.59 | 0.87 | 0.73 | 0.92 | 0.72 | 0.88 | ↑**92.33** |

(1) *Encoders&Dictionary*: We start by comparing three implementations: FDA on text encoders, fusion encoders, and both, denoted as $\mathcal{T}$, $\mathcal{H}$, and $\mathcal{T}\&\mathcal{H}$. As presented in the top rows of Table.5, we find that $\mathcal{T}$ performs the worst among all, indicating that an early subtraction is insufficient for removing such subtraction. Although $\mathcal{T}\&\mathcal{H}$ provides a significant robustness boost, it costs an evident 2% performance drop on performance, implying that subtraction on both encoders is too strong and potentially causes contextual distortion. $\mathcal{H}$ performs the best as it helps models concentrate while preserving the contextual meaning.

For the dictionary, we use the off-the-shelf stopwords dictionary in (Li et al., 2020), containing 208 words/symbols, denoted as Full Dict. Furthermore, we use a shortlisted dictionary, by only using the most commonly used function words, containing 91 crucial function words, denoted as Shortlisted Dict. Both dictionary settings are trained with FDA $L^{all}$ to maximize their impacts on training. As presented in the lower row of Table.5, there are no significant performance gaps between the two settings, with Full Dict performing slightly worse regarding both clean and adversarial examples. We attribute the minor degradation to the length of the stopwords dictionary, which could unnecessarily skim words and distort the context. We provide the shortlisted dictionary in Sec.F of the Appendix.

(2) *Attention Head &Layer:* We then investigate the index of the layers $L$ and attention heads $H$ for retrieval and grounding, as presented in Table.6 and Table.7. Specifically, we train a series models using FDA but using different $L$ and $H$: for layers, we use all, 0-1, and 0 layers, denoted as $L^{all}, L^{0-1}, L^{0}$; for attention heads, we use all heads, 1st half (0-5) and the second half (6-11), denoted as $H^{all}, H^{0-5}, H^{6-11}$. For T2IR/I2TR, as shown in Table.6, we find that the shallow implementations of FDA, i.e., $L^{0}/L^{0-1}, H^{0-5}$ consistently yield the best performance on robustness on **all models**. Specifically, $L^{0}, H^{0-5}$ constantly achieves the best clean performance, leading other counterparts by 0.1~0.2%. We further test the leading 3 settings on retrieval tasks, i.e., $L^{all}/L^{0}/L^{0-1}, H^{0-5}$ on VG. As shown in 7, we find the shallow $L^{0-1}, H^{0-5}$ settings still top w.r.t. both adversarial and clean examples, leading other settings by 10-20%/0.3-0.4%, respectively. Overall, FDA remains solid and insensitive to the head/layer parameters.

**FDA v.s. Masking & Other DA.** We start by providing comparisons of our FDA and fine-tuning models by directly masking function words, including content words and nouns for sanity checks.

For variants of FDA, we further compare our FDA with Adjective DA (ADA) and Determiner DA (DDA). Specifically, we choose determiners (DET) and adjectives because DET indicates using a small subset (i.e., a/an/the) of function words, while ADJ adopts a completely different set of words. Results are presented in Table.8. Note: *We only test on PGD and APGD since MAPGD is not applicable for nouns and content words.*

Overall, masking content words and nouns yields the largest performance drop, making it inviable for robustness evaluation. This aligns with the intuition that these words carry extensive semantic information crucial for VLM tasks. Furthermore, naively masking function words leads to evident performance drop (∼1%) and brings negligible robustness. As a subset of FDA, DDA has identical clean performance but much less robustness. ADA also shows subpar performance compared with FDA. In sum, **FDA leads the clean and adversarial performance among other variants**.

**Zero-shot Performance.** We also adopt three different head/layer settings of FDA for zero-shot T2IR/I2TR/VG on ALBEF and BLIP with $H^{0-5}$. Results are presented in Table.9. We find that $L^{all}$ performs the best for all tasks and all models. This not only suggests that $L^{all}$ serves as the most generalizable setting for multiple VL tasks and models, but also implies the feasibility of FDA for performance boost on zero-shot tasks.

*Takeaway:* Implementing FDA on the fusion encoder would be an ideal practice for all models. While we do not rule out potentially better performance by using different $L/H$, one can adopt the setting in the paper on similar downstream tasks for stability, or simply use $L^{all}, H^{0-5}$ as a safe choice.

## 4.4 ANALYSIS AND VISUALIZATION

We notice that the results of APGD and MAPGD somewhat worsen after adversarial fine-tuning, e.g., TeCoA and FARE on ALBEF, FARE on BLIP in Table.1, etc. As previously illustrated in Fig.2, defending against targeted attacks requires a more aligned vision-language embedding. Consequently, we hypothesize that such abnormality potentially originates from the disruption in vision-language alignment brought by adversarial noise for enhanced robustness.

To validate our speculation, we visualize the vision-language distribution of ALBEF to-

Table 8: Comparison for directly removing content words (CONT), nouns (NOUN), function words (FUNC), Determiner DA (DDA), and Adjective DA (ADA). $\Delta_{ASR}$ is presented using the average results for PGD and APGD.**FDA shows significant advantages over directly masking and other de-attention variants.**

| Maksed | Clean (R@1) (↑) | | Avg ASR Drop |
|---|---|---|---|
| Words | T2IR | I2TR | $\Delta_{ASR}$ (↑) |
| N/A | 95.90 | 85.60 | - |
| CONT | 21.50 | 11.10 | - |
| NOUN | 68.60 | 44.62 | - |
| FUNC | 94.00 | 80.86 | ↑ 1.56 |
| DDA | 95.60 | 85.42 | ↑ 9.28 |
| ADA | 95.50 | 85.38 | ↑ 15.10 |
| FDA | **95.60** | **85.50** | **↑ 23.07** |

Table 9: Zero-shot performance by applying FDA as a plug-and-play tool on T2IR/I2TR on ALBEF/BLIP and VG on ALBEF. T2IR/I2TR uses R@1/5/10, while VG uses accuracies.

| Tasks | Models | Method | Avg Performance | |
|---|---|---|---|---|
| Retrieval | ALBEF | w/o FDA | 92.01 | - |
| | | $L^0$ | 92.02 | ↑ 0.01 |
| | | $L^{0-1}$ | **92.41** | ↑ 0.40 |
| | | $L^{all}$ | 92.17 | ↑ 0.16 |
| | BLIP | w/o FDA | 92.24 | - |
| | | $L^0$ | 92.19 | ↓ 0.05 |
| | | $L^{0-1}$ | 92.22 | ↓ 0.02 |
| | | $L^{all}$ | **92.71** | ↑ 0.47 |
| VG | ALBEF | w/o FDA | 53.12 | - |
| | | $L^0$ | 52.72 | ↓ 0.40 |
| | | $L^{0-1}$ | 52.68 | ↓ 0.44 |
| | | $L^{all}$ | **53.34** | ↑ 0.22 |

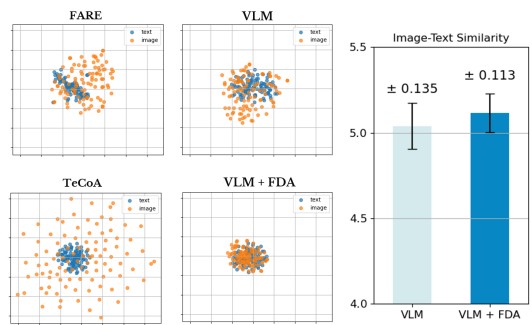

Figure 3: *Left*: T-SNE of the vision-language embedding of vanilla VLM, FDA, FARE, and TeCoA. **Our FDA is the most aligned model**. *Right*: Comparison of text-image similarity for vanilla VLM versus VLM + FDA. **Our FDA yields tightened alignment with larger similarities and smaller variances.**

gether with TeCoA, FARE, and FDA, as shown in the left graph Fig.3. From the left graph, we find that both FARE and TeCoA (left column) yield a severely disrupted embedding, where images and texts sparsely scatter away from each other. On the other hand, our FDA (lower right) has the most aligned cross-modal embedding, as all images and texts remain tightly aligned with each other. To numerically compare the alignment of FDA and the vanilla model, we record the top 200 average white-box text-image similarity scores. As shown in the right figure of Fig.3, applying FDA generates higher average text-image similarity scores and lower variations.

## 4.5 Limitation

Although direct subtraction introduces superior efficiency, FDA could be potentially improved through a modular or algorithmic approach for more refined removal. Furthermore, we did not implement FDA to fine-tune a larger VLM or verify the effectiveness of FDA using PEFT methods such as LoRa (Hu et al., 2022) due to the hardware limitation. Additionally, our FDA is designed and implemented on backbones featuring a fusion encoder, but not been applied to projector-based models such as LLaVA (Liu et al., 2023) series and Qwen(Bai et al., 2023; Team et al., 2024; Bai et al., 2025; Yang et al., 2025) series, etc. Nevertheless, FDA could be seamlessly adopted on these backbones, which is a valuable exploration for future work. Finally, although we have built a shortlisted dictionary based on existing ones, it is possible that the optimal dictionary might have less/more words, which requires enormous effort for validation.

## 5 Conclusion

In this paper, we propose Function-word De-Attention (FDA) calculates the original and the function-word cross-attention within attention heads, and differentially subtracts the latter from the former for more aligned and robust VLMs. Specifically, we tested the FDA on 2 downstream tasks, 3 datasets, and 3 models, and evaluated all methods under 6 attacks. By comparing with existing SOTA defenses, our FDA shows superiority of FDA in boosting robustness and clean performance. We also provide an in-depth analysis of FDA and validate its boost on zero-shot performance. Our work validate the redundancy of function words in vision-language models and the subsequent vulnerability brought by these words, calling for more research interest in further exploration.

## Acknowledgement

This work was supported by National Key Research and Development Program of China (2023YFB3107401), the National Natural Science Foundation of China (T2341003, 62521002, U2441240, U24B20185, 62376210, 62132011, 62406240, U244120060).

## Ethics Statement

We acknowledge that all authors of our papers are required to read the Code of Ethics, adhere to it, and explicitly acknowledge this during the submission process. Contribute to society and to human well-being. All authors: i) uphold high standards of scientific excellence; ii) avoid harm; iii) be honest, trustworthy, and transparent; iv) be fair and take action to avoid discrimination; v) respect the work required to produce new ideas and artefacts; vi) respect privacy; vii) honour confidentiality.

## Reproducibility Statement

The detailed information about the implementation of FDA is provided in Section.4. The data used in this paper is open-source, and the details/full results are stated in the Appendix. The code and the checkpoint will be publicly available along with sufficient instructions to faithfully reproduce the main experimental results/visualization, and detailed instructions to transfer to other backbones.

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

# Appendix for Pay Less Attention to Function Words for Free Robustness of Vision-Language Models

## A  FDA v.s. Adaptive Words selection

We further qualitatively verify FDA with 3 adaptive selection methods (i.e., using per-token similarity of text and image features to choose tokens with low similarity): i) setting threshold of $\mu - \delta$; ii) setting threshold of $\mu - 2\delta$, with $\mu, \delta$ being the mean and std of the text-image similarity. We further choose the lowest N tokens, with N being the number of function words in the texts. We denote them as SIM-$\delta$, SIM-$2\delta$, and SIM-N, respectively. Furthermore, we **record the % of selected words that are in our shortlisted function words dictionary**. Results are shown in Table.10.

In summary, we find that **the gained robustness of VLMs increased as the proportion of function words increased**. While we cannot design an adaptive mechanism that perfectly aligns with using the function words dictionary, we find that while the vulnerability of VLMs does not necessarily come from low-similarity (or low semantic) words, **there is an evident correlation between the percentage of function words and the gained robustness**.

## B  Details for attacks and evaluation metrics

We first introduce the attacks and evaluation metrics for each VL task, including the scenarios where a targeted attack is considered successful and the corresponding metrics.

**Text-to-Image/Image-to-Text Retrieval.** For T2IR, a successful targeted attack is only when the manipulated images emerge in the Top 1/5 position given the targeted text queries; for I2TR, a successful attack is only when the targeted texts emerge in the Top 1/5 position given the manipulated images as the query. Consequently, the ASR of T2IR/I2TR would be the hit rate at the top 1/5, i.e., the probability of appearance in the top 1/5 position, denoted as ASR@1/5. In the main paper, we use the average of ASR@1/5 as the overall ASR. Untargeted attacks follow the identical setting of existing works, i.e., lowering the R@1/5 of the victim models.

**Visual Grounding.** For visual grounding, we choose to obfuscate the model by fooling it into recognizing other objects as the target, or, if there is only one object in the image, locating the position of the object incorrectly (top-left corner). A successful attack is when the IOU of the targeted bounding box and the model bounding box is larger than 0.5, i.e., the model locates the object within the targeted bounding box. As for untargeted attacks, we follow existing settings to lower the accuracy of the victim model and calculate the drops as ASR.

## C  Full results for targeted attacks

In this section, we provide full results for all targeted attacks on all models and tasks. Specifically, for T2IR and I2TR, results on ALBEF is given in Table.11 and Table.12, results on TCL is given in Table.13 and Table.14, and results on BLIP is given in Table.15 and Table.16, respectively. Targeted attacks for visual grounding on ALBEF are given in Table.17.

## D  Full results for untargeted attacks

In this section, we provide full results for all untargeted attacks on all models and tasks. Specifically, for T2IR and I2TR, results on AL-BEF is given in Table.18 and Table.19, and

Table 10: Comparison between FDA and adaptive selection, i.e., using image-text similarity to choose less informative tokens. SIM-$\delta$/-$2\delta$ indicates using $\mu - \delta$ and $\mu - 2\delta$ as the de-attention threshold, with $\mu, \delta$ being the mean and std of the text-image similarity. SIM-N refers to choosing the lowest N tokens, with N being the number of function words in the text. **% of words** means the percentage of function words in the selected ones. **Results confirm the correlation between the proportion of function words and the gained robustness.**

| Defense | % of Words in Dictionary | Clean R@1 (↑) | | ASR Drop $\Delta_{ASR}$ (↑) | | |
|---|---|---|---|---|---|---|
| | | T2IR | I2TR | $2/255$ | $4/255$ | Avg |
| **SIM-N** | 25.95 | **95.90** | **85.50** | 2.44 | 12.81 | ↑ 7.62 |
| **SIM-$2\delta$** | 74.53 | 95.60 | 85.32 | 9.38 | 13.86 | ↑ 8.39 |
| **SIM-$\delta$** | 79.49 | 95.30 | 85.38 | 12.85 | 11.54 | ↑ 12.41 |
| **FDA** | 100.00 | **95.90** | **85.50** | **27.61** | **18.53** | ↑ **23.07** |

Table 11: ASR of white-box *targeted* attacks against **Text-to-Image Retrieval** on Flickr30k and COCO. The model is **ALBEF**. Changes over unattacked values are presented in parentheses. All results are in percentage (%). ASR@1/5 indicates the attack success rate of the adversarial image showing up in the top-1/5 position of the targeted text queries.

| Dataset | $l_\infty$ | Defense | Clean↑ | PGD | | APGD | | MAPGD | |
|---|---|---|---|---|---|---|---|---|---|
| | | | | ASR@1↓ | ASR@5↓ | ASR@1↓ | ASR@5↓ | ASR@1↓ | ASR@5↓ |
| Flickr | 2/255 | No Defense | 95.90 | 0.30 (+0.20) | 7.50 (+6.50) | 14.60 (+14.50) | 15.70 (+14.70) | 50.10 (+50.00) | 81.90 (+80.90) |
| | | TeCoA | 91.20 | 0.20 (+0.20) | 5.30 (+4.90) | 18.70 (+18.70) | 20.40 (+20.00) | 59.90 (+59.90) | 86.40 (+86.00) |
| | | FARE | 91.10 | 0.10 (+0.10) | 5.10 (+4.80) | 17.20 (+17.20) | 17.80 (+17.30) | 58.10 (+57.90) | 80.50 (+80.00) |
| | | FDA-$L^0$ | 95.60 | 0.10 (+0.10) | 7.30 (+6.60) | 12.10 (+12.10) | 13.40 (+12.70) | 43.60 (+43.60) | 73.90 (+73.20) |
| | | FDA-$L^{0-1}$ | 95.40 | 0.20 (+0.20) | 6.90 (+6.10) | 12.00 (+12.00) | 12.80 (+12.00) | 43.30 (+43.30) | 73.60 (+72.80) |
| | | FDA-$L^{all}$ | 95.40 | 0.40 (+0.40) | 5.90 (+5.20) | 12.80 (+12.80) | 14.20 (+13.50) | 43.50 (+43.50) | 77.30 (+76.60) |
| | 4/255 | No Defense | 95.90 | 4.30 (+4.20) | 14.10 (+13.10) | 16.50 (+16.40) | 16.60 (+15.60) | 75.00 (+74.90) | 87.00 (+86.00) |
| | | TeCoA | 91.20 | 3.90 (+3.90) | 14.70 (+14.30) | 19.40 (+19.40) | 19.60 (+19.20) | 81.00 (+81.00) | 90.00 (+89.60) |
| | | FARE | 91.10 | 4.00 (+4.00) | 14.80 (+14.50) | 18.80 (+18.80) | 18.80 (+18.30) | 79.70 (+79.70) | 85.90 (+85.40) |
| | | FDA-$L^0$ | 95.60 | 2.90 (+2.90) | 13.50 (+12.80) | 13.90 (+13.90) | 14.10 (+13.40) | 69.00 (+69.00) | 82.40 (+81.70) |
| | | FDA-$L^{0-1}$ | 95.40 | 3.00 (+3.00) | 12.40 (+11.60) | 13.90 (+13.90) | 14.00 (+13.20) | 68.10 (+68.10) | 81.30 (+80.50) |
| | | FDA-$L^{all}$ | 95.40 | 3.00 (+3.00) | 13.50 (+12.80) | 14.60 (+14.60) | 14.80 (+14.10) | 68.90 (+68.90) | 84.10 (+83.40) |
| COCO | 2/255 | No Defense | 77.60 | 0.22 (+0.18) | 1.80 (+1.72) | 10.18 (+10.14) | 11.94 (+11.86) | 20.14 (+20.14) | 40.88 (+0.80) |
| | | TeCoA | 68.04 | 0.10 (+0.08) | 0.66 (+0.56) | 16.42 (+16.40) | 17.40 (+17.34) | 24.52 (+24.50) | 44.02 (+43.92) |
| | | FARE | 69.28 | 0.08 (+0.06) | 0.55 (+0.45) | 19.64 (+19.62) | 25.82 (+25.72) | 21.76 (+21.74) | 43.74 (+43.64) |
| | | FDA-$L^{0-1}$ | 77.70 | 0.26 (+0.22) | 1.58 (+1.46) | 9.00 (+ 8.98) | 10.40 (+10.30) | 18.28 (+18.26) | 37.00 (+36.90) |
| | 4/255 | No Defense | 77.60 | 2.10 (+2.06) | 7.44 (+7.36) | 14.26 (+14.22) | 14.80 (+14.72) | 43.74 (+43.70) | 58.72 (+58.64) |
| | | TeCoA | 68.04 | 0.52 (+0.50) | 2.74 (+2.64) | 25.00 (+24.98) | 26.46 (+26.36) | 53.56 (+53.54) | 66.28 (+66.18) |
| | | FARE | 69.28 | 0.40 (+0.38) | 2.52 (+2.42) | 30.94 (+30.92) | 33.82 (+33.72) | 54.46 (+54.44) | 72.48 (+72.38) |
| | | FDA-$L^{0-1}$ | 77.70 | 1.74 (+1.70) | 6.06 (+5.94) | 11.76 (+11.74) | 12.08 (+11.98) | 37.92 (+37.90) | 51.98 (+51.88) |

Table 12: ASR of white-box *targeted* attacks against **Image-to-Text Retrieval** on Flickr30k and COCO. The model is **ALBEF**. Changes over unattacked values are presented in parentheses. All results are in percentage (%). ASR@1/5 indicates the attack success rate of the targeted text queries showing up in the top-1/5 position of the adversarial image.

| Dataset | $l_\infty$ | Defense | Clean↑ | PGD | | APGD | | MAPGD | |
|---|---|---|---|---|---|---|---|---|---|
| | | | | ASR@1↓ | ASR@5↓ | ASR@1↓ | ASR@5↓ | ASR@1↓ | ASR@5↓ |
| Flickr | 2/255 | No Defense | 85.60 | 0.30 (+0.30) | 1.10 (+1.10) | 14.40 (+14.40) | 15.40 (+15.40) | 53.50 (+53.50) | 63.50 (+63.50) |
| | | TeCoA | 81.44 | 0.10 (+0.08) | 1.00 (+1.00) | 16.42 (+16.40) | 17.40 (+17.34) | 56.90 (+56.90) | 65.70 (+65.70) |
| | | FARE | 81.48 | 0.10 (+0.10) | 1.00 (+1.00) | 19.64 (+19.62) | 25.82 (+25.72) | 56.30 (+56.30) | 65.90 (+65.90) |
| | | FDA-$L^0$ | 85.50 | 0.10 (+0.10) | 0.60 (+0.60) | 12.30 (+12.30) | 12.80 (+12.80) | 46.50 (+46.50) | 56.20 (+56.20) |
| | | FDA-$L^{0-1}$ | 85.32 | 0.10 (+0.10) | 0.80 (+0.80) | 12.10 (+12.10) | 12.50 (+12.50) | 46.90 (+46.90) | 55.90 (+55.90) |
| | | FDA-$L^{all}$ | 85.40 | 0.20 (+0.20) | 1.00 (+1.00) | 13.50 (+13.50) | 13.70 (+13.70) | 48.50(+48.50) | 58.00 (+58.00) |
| | 4/255 | No Defense | 85.60 | 4.50 (+4.50) | 9.80 (+9.80) | 15.70 (+15.70) | 15.90 (+15.90) | 74.40 (+74.40) | 79.00 (+79.00) |
| | | TeCoA | 81.44 | 2.80 (+2.80) | 6.40 (+6.40) | 18.30 (+18.30) | 18.60 (+18.60) | 78.10 (+78.10) | 81.10 (+81.10) |
| | | FARE | 81.48 | 3.40 (+3.40) | 7.00 (+7.00) | 18.30 (+18.30) | 18.40 (+18.40) | 77.00 (+77.00) | 80.60 (+80.60) |
| | | FDA-$L^0$ | 85.50 | 3.30 (+3.30) | 6.50 (+6.50) | 13.70 (+13.70) | 13.70 (+13.70) | 68.80 (+68.80) | 72.40 (+72.40) |
| | | FDA-$L^{0-1}$ | 85.32 | 3.10 (+3.10) | 6.90 (+6.90) | 13.40 (+13.40) | 13.50 (+13.50) | 69.30 (+69.30) | 72.30 (+72.30) |
| | | FDA-$L^{all}$ | 85.40 | 4.00 (+4.00) | 7.80 (+7.80) | 14.10 (+14.10) | 14.20 (+14.20) | 72.20 (+72.20) | 75.20 (+75.20) |
| COCO | 2/255 | No Defense | 60.70 | 0.22 (+0.22) | 0.50 (+0.48) | 7.68 (+7.68) | 10.04 (+10.02) | 14.64 (+14.64) | 24.16 (+21.14) |
| | | TeCoA | 53.07 | 0.02 (+0.02) | 0.12 (+0.12) | 10.82 (+10.82) | 13.58 (+13.54) | 14.32 (+14.32) | 33.74 (+33.74) |
| | | FARE | 53.58 | 0.00 (+0.00) | 0.06 (+0.04) | 11.80 (+11.80) | 17.40 (+17.38) | 21.62 (+12.62) | 20.92 (+20.90) |
| | | FDA-$L^{0-1}$ | 60.63 | 0.16 (+0.10) | 0.42 (+0.40) | 7.14 (+ 7.14) | 12.34 (+12.32) | 16.72 (+16.72) | 26.66 (+26.64) |
| | 4/255 | No Defense | 60.70 | 1.46 (+1.46) | 3.38 (+3.36) | 11.11 (+11.10) | 13.26 (+13.24) | 50.10 (+50.00) | 81.90 (+80.90) |
| | | TeCoA | 53.07 | 0.16 (+0.16) | 0.56 (+0.56) | 18.58 (+18.58) | 25.56 (+25.56) | 38.42 (+38.42) | 52.00 (+51.96) |
| | | FARE | 53.58 | 0.22 (+0.22) | 0.50 (+0.48) | 21.84 (+21.84) | 26.88 (+26.86) | 31.94 (+31.94) | 47.30 (+47.28) |
| | | FDA-$L^{0-1}$ | 60.63 | 1.20 (+1.20) | 2.92 (+2.90) | 9.88 (+ 9.88) | 11.28 (+11.26) | 27.74(+27.74) | 37.94 (+37.92) |

results on BLIP is given in Table.20 and Table.21, respectively. Untargeted attacks for visual grounding on ALBEF are given in Table.22.

# E    FULL RESULTS FOR ABLATION STUDIES

In this section, we provide full results for all ablation studies. T2IR and I2TR results are given in Table.23 and Table.24. Zero-shot performance is given in Table.25.

Table 13: ASR of white-box *targeted* attacks against **Text-to-Image Retrieval** on Flickr30k. The model is **TCL**. Changes over unattacked values are presented in parentheses. All results are in percentage (%). ASR@1/5 indicates the attack success rate of the adversarial image showing up in the top-1/5 position of the targeted text queries.

| $l_\infty$ | Defense | Clean ↑ | PGD | | APGD | | MAPGD | |
|---|---|---|---|---|---|---|---|---|
| | | | ASR@1 ↓ | ASR@5 ↓ | ASR@1 ↓ | ASR@5 ↓ | ASR@1 ↓ | ASR@5 ↓ |
| 2/255 | No Defense | 94.90 | 2.10 (+2.10) | 18.80 (+18.40) | 66.00 (+66.00) | 75.20 (+74.80) | 50.90 (+50.90) | 82.50 (+82.10) |
| | TeCoA | 92.10 | 3.10 (+3.10) | 19.30 (+19.00) | 61.00 (+61.00) | 71.70 (+71.40) | 44.20 (+44.20) | 74.10 (+73.80) |
| | FARE | 91.70 | 3.20 (+3.20) | 20.40 (+20.20) | 63.10 (+63.10) | 71.90 (+71.70) | 46.80 (+46.80) | 75.20 (+75.00) |
| | FDA-$L^0$ | 94.40 | 1.90 (+1.90) | 15.40 (+15.10) | 44.90 (+44.90) | 52.00 (+51.70) | 52.40 (+52.40) | 84.60 (+84.30) |
| | FDA-$L^{0-1}$ | 94.20 | 2.40 (+2.40) | 17.00 (+16.60) | 46.30 (+46.30) | 55.80 (+55.00) | 54.40 (+54.40) | 86.70 (+85.90) |
| | FDA-$L^{all}$ | 94.10 | 2.30 (+2.30) | 16.80 (+16.40) | 50.80 (+50.80) | 60.90 (+60.50) | 54.90 (+54.90) | 87.10 (+86.70) |
| 4/255 | No Defense | 94.90 | 21.50 (+21.50) | 54.00 (+53.60) | 80.30 (+80.30) | 82.00 (+81.60) | 75.20 (+75.20) | 88.10 (+87.70) |
| | TeCoA | 92.10 | 27.80 (+27.80) | 60.90 (+60.60) | 79.70 (+79.70) | 81.60 (+81.30) | 74.10 (+74.10) | 86.10 (+85.80) |
| | FARE | 91.70 | 30.20 (+30.20) | 62.30 (+62.10) | 80.30 (+80.30) | 81.80 (+81.60) | 73.20 (+73.20) | 86.10 (+85.90) |
| | FDA-$L^0$ | 94.40 | 17.90 (+17.90) | 43.00 (+42.70) | 56.80 (+56.80) | 60.60 (+60.30) | 80.20 (+80.20) | 92.30 (+92.00) |
| | FDA-$L^{0-1}$ | 94.20 | 18.80 (+18.80) | 44.90 (+44.50) | 60.20 (+60.20) | 64.50 (+63.70) | 80.40 (+80.40) | 92.90 (+92.10) |
| | FDA-$L^{all}$ | 94.10 | 19.00 (+19.00) | 44.90 (+44.50) | 66.10 (+66.10) | 69.10 (+68.70) | 79.80 (+79.80) | 94.50 (+94.10) |

Table 14: ASR of white-box *targeted* attacks against **Image-to-Text Retrieval** on Flickr30k. The model is **TCL**. Changes over unattacked values are presented in parentheses. All results are in percentage (%). ASR@1/5 indicates the attack success rate of the targeted text queries showing up in the top-1/5 position of the adversarial image.

| $l_\infty$ | Defense | Clean ↑ | PGD | | APGD | | MAPGD | |
|---|---|---|---|---|---|---|---|---|
| | | | ASR@1 ↓ | ASR@5 ↓ | ASR@1 ↓ | ASR@5 ↓ | ASR@1 ↓ | ASR@5 ↓ |
| 2/255 | No Defense | 84.02 | 2.50 (+2.50) | 5.70 (+5.70) | 64.10 (+64.10) | 66.80 (+66.80) | 50.70 (+50.70) | 58.90 (+58.90) |
| | TeCoA | 80.40 | 2.10 (+2.10) | 6.10 (+6.10) | 69.60 (+69.60) | 72.00 (+72.00) | 65.50 (+65.50) | 70.40 (+70.40) |
| | FARE | 78.22 | 3.10 (+3.10) | 6.10 (+6.10) | 59.70 (+59.70) | 62.80 (+62.80) | 65.20 (+65.20) | 69.50 (+69.50) |
| | FDA-$L^0$ | 83.82 | 1.90 (+1.90) | 5.30 (+5.30) | 43.40 (+43.40) | 45.60 (+45.60) | 52.90 (+52.90) | 62.10 (+62.10) |
| | FDA-$L^{0-1}$ | 83.96 | 1.80 (+1.80) | 4.80 (+4.80) | 45.30 (+45.30) | 47.50 (+47.50) | 53.90 (+53.90) | 64.40 (+64.40) |
| | FDA-$L^{all}$ | 83.98 | 1.30 (+1.30) | 4.60 (+4.60) | 51.50 (+51.50) | 54.30 (+54.30) | 56.00 (+56.00) | 64.40 (+64.40) |
| 4/255 | No Defense | 84.02 | 24.60 (+24.60) | 34.70 (+34.70) | 78.80 (+77.80) | 78.60 (+78.60) | 70.40 (+70.40) | 75.50 (+75.50) |
| | TeCoA | 80.40 | 29.60 (+29.60) | 34.70 (+34.70) | 76.00 (+76.00) | 77.20 (+77.20) | 65.50 (+65.50) | 70.40 (+70.40) |
| | FARE | 78.22 | 33.30 (+33.30) | 39.50 (+39.50) | 76.20 (+76.20) | 77.70 (+77.70) | 65.20 (+65.20) | 69.50 (+69.50) |
| | FDA-$L^0$ | 83.82 | 20.00 (+20.00) | 28.50 (+28.50) | 56.10 (+56.10) | 56.90 (+56.90) | 75.50 (+75.50) | 80.10 (+80.10) |
| | FDA-$L^{0-1}$ | 83.96 | 20.10 (+20.10) | 29.30 (+29.30) | 60.00 (+60.00) | 60.80 (+60.80) | 77.80 (+77.80) | 82.70 (+82.70) |
| | FDA-$L^{all}$ | 83.98 | 22.20 (+22.20) | 31.10 (+31.10) | 64.40 (+64.40) | 65.50 (+65.50) | 79.50 (+79.50) | 84.10 (+84.10) |

Table 15: ASR of white-box *targeted* attacks against **Text-to-Image Retrieval** on Flickr30k. The model is **BLIP**. Changes over unattacked values are presented in parentheses. All results are in percentage (%). ASR@1/5 indicates the attack success rate of the adversarial image showing up in the top-1/5 position of the targeted text queries.

| $l_\infty$ | Defense | Clean ↑ | PGD | | APGD | | MAPGD | |
|---|---|---|---|---|---|---|---|---|
| | | | ASR@1 ↓ | ASR@5 ↓ | ASR@1 ↓ | ASR@5 ↓ | ASR@1 ↓ | ASR@5 ↓ |
| 2/255 | No Defense | 97.20 | 2.50 (+2.50) | 46.10 (+46.10) | 80.50 (+81.10) | 75.20 (+74.80) | 57.90 (+57.90) | 84.70 (+84.30) |
| | TeCoA | 81.50 | 4.30 (+4.30) | 16.20 (+16.20) | 19.30 (+19.30) | 45.70 (+45.60) | 14.80 (+14.80) | 39.20 (+39.10) |
| | FARE | 79.40 | 1.00 (+1.00) | 10.30 (+10.10) | 12.90 (+12.90) | 46.20 (+46.20) | 9.70 (+ 9.70) | 38.00 (+37.90) |
| | FDA-$L^0$ | 96.80 | 3.10 (+3.10) | 12.00 (+11.90) | 13.60 (+13.60) | 26.30 (+26.20) | 24.20 (+24.20) | 61.70 (+61.60) |
| | FDA-$L^{0-1}$ | 96.50 | 3.00 (+3.00) | 12.40 (+12.30) | 12.30 (+12.30) | 25.60 (+25.70) | 22.10 (+22.10) | 59.90 (+59.80) |
| | FDA-$L^{all}$ | 96.50 | 3.20 (+3.20) | 42.00 (+41.80) | 16.00 (+16.00) | 39.60 (+39.40) | 24.50 (+24.50) | 66.30 (+66.30) |
| 4/255 | No Defense | 97.20 | 31.80 (+31.80) | 90.60 (+90.20) | 79.80 (+79.80) | 93.00 (+92.60) | 57.90 (+57.90) | 84.70 (+84.30) |
| | TeCoA | 81.50 | 46.20 (+46.20) | 73.00 (+72.90) | 60.40 (+60.40) | 83.10 (+83.00) | 49.50 (+49.50) | 74.80 (+74.80) |
| | FARE | 79.40 | 23.90 (+23.90) | 61.50 (+61.30) | 42.30 (+42.30) | 82.70 (+82.60) | 33.90 (+33.90) | 74.70 (+74.70) |
| | FDA-$L^0$ | 96.80 | 13.60 (+13.60) | 19.80 (+19.70) | 16.40 (+16.40) | 43.60 (+43.50) | 44.70 (+44.70) | 78.50 (+78.40) |
| | FDA-$L^{0-1}$ | 96.50 | 13.20 (+13.20) | 18.60 (+18.50) | 15.10 (+15.10) | 41.60 (+41.70) | 43.40 (+43.40) | 77.90 (+77.80) |
| | FDA-$L^{all}$ | 96.50 | 31.60 (+31.60) | 86.80 (+86.60) | 21.40 (+21.40) | 62.00 (+61.80) | 50.00 (+50.00) | 81.90 (+81.70) |

Table 16: ASR of white-box *targeted* attacks against **Image-to-Text Retrieval** on Flickr30k. The model is **BLIP**. Changes over unattacked values are presented in parentheses. All results are in percentage (%). ASR@1/5 indicates the attack success rate of the targeted text queries showing up in the top-1/5 position of the adversarial image.

| $l_\infty$ | Defense | Clean ↑ | PGD ASR@1 ↓ | PGD ASR@5 ↓ | APGD ASR@1 ↓ | APGD ASR@5 ↓ | MAPGD ASR@1 ↓ | MAPGD ASR@5 ↓ |
|---|---|---|---|---|---|---|---|---|
| 2/255 | No Defense | 87.30 | 7.00 (+7.00) | 16.60 (+16.60) | 54.90 (+54.90) | 65.00 (+65.00) | 37.60 (+37.60) | 50.90 (+50.90) |
| | TeCoA | 68.00 | 2.40 (+2.40) | 5.90 (+5.90) | 13.90 (+13.90) | 25.10 (+25.10) | 9.30 (+9.30) | 19.80 (+19.80) |
| | TeCoA + FDA-$L^{0-1}$ | 67.78 | 2.20 (+2.20) | 6.20 (+6.20) | 12.90 (+12.90) | 25.40 (+25.40) | 8.90 (+8.90) | 19.60 (+19.60) |
| | FARE | 65.64 | 0.40 (+0.40) | 1.90 (+1.90) | 11.30 (+11.30) | 19.60 (+19.60) | 8.00 (+8.00) | 15.10 (+15.10) |
| | FARE + FDA-$L^{0-1}$ | 66.22 | 0.40 (+0.40) | 2.10 (+2.10) | 9.40 (+9.40) | 17.50 (+17.50) | 6.00 (+6.00) | 13.60 (+13.60) |
| | FDA-$L^0$ | 86.86 | 4.40 (+4.40) | 5.30 (+5.30) | 14.10 (+14.10) | 15.70 (+15.70) | 31.70 (+31.70) | 41.60 (+41.60) |
| | FDA-$L^{0-1}$ | 86.86 | 4.60 (+4.60) | 4.80 (+4.80) | 13.10 (+13.10) | 14.40 (+14.40) | 30.60 (+30.60) | 39.40 (+39.40) |
| | FDA-$L^{all}$ | 86.94 | 6.70 (+3.20) | 14.60 (+14.60) | 16.90 (+16.90) | 18.60 (+18.60) | 35.00 (+35.00) | 46.50 (+46.50) |
| 4/255 | No Defense | 84.02 | 58.80 (+58.80) | 74.90 (+74.90) | 83.70 (+83.70) | 88.10 (+88.10) | 67.00 (+67.00) | 75.90 (+75.90) |
| | TeCoA | 68.00 | 46.60 (+46.60) | 58.10 (+58.10) | 58.90 (+58.90) | 69.00 (+69.00) | 47.10 (+47.10) | 60.10 (+60.10) |
| | TeCoA + FDA-$L^{0-1}$ | 67.78 | 44.50 (+44.50) | 59.10 (+59.10) | 59.10 (+59.10) | 70.30 (+70.30) | 47.40 (+47.40) | 61.10 (+61.10) |
| | FARE | 65.64 | 23.90 (+23.90) | 37.30 (+37.30) | 45.70 (+45.70) | 60.90 (+60.90) | 34.00 (+34.00) | 50.90 (+50.90) |
| | FARE + FDA-$L^{0-1}$ | 66.22 | 24.70 (+24.70) | 37.10 (+37.10) | 43.90 (+43.90) | 58.60 (+58.60) | 31.90 (+31.90) | 48.20 (+48.20) |
| | FDA-$L^0$ | 86.86 | 14.80 (+14.80) | 15.30 (+15.30) | 16.80 (+16.80) | 17.30 (+17.30) | 53.40 (+53.40) | 60.70 (+60.70) |
| | FDA-$L^{0-1}$ | 86.86 | 14.20 (+14.20) | 14.70 (+14.70) | 16.20 (+16.20) | 16.40 (+16.40) | 52.00 (+52.00) | 59.00 (+59.00) |
| | FDA-$L^{all}$ | 86.94 | 55.20 (+52.20) | 70.80 (+70.80) | 21.80 (+21.80) | 22.10 (+22.10) | 60.20 (+60.20) | 68.70 (+68.70) |

Table 17: Attack success rate (ASR) of *targeted* PGD/APGD/MAPGD (masked APGD) against for **Visual Grounding** (VG) on RefCOCO+. All results are presented in percentage (%). Changes over unattacked values are presented in parentheses.

| $l_\infty$ | Defense | Clean Performance Val_d | Test A | Test B | Test A Split (↓) PGD | APGD | MAPGD | Test B Split (↓) PGD | APGD | MAPGD |
|---|---|---|---|---|---|---|---|---|---|---|
| 2/255 | No Defense | 58.50 | 65.90 | 46.30 | 16.40 (+6.00) | 20.40 (+10.00) | 20.40 (+10.00) | 25.73 (+4.80) | 26.53 (+5.60) | 27.30 (+6.37) |
| | TeCoA | 57.20 | 64.70 | 45.00 | 17.87 (+6.00) | 18.67 (+6.80) | 18.93 (+7.06) | 24.00 (+2.67) | 26.27 (+4.94) | 26.13 (+4.80) |
| | FARE | 56.40 | 64.20 | 44.70 | 18.40 (+5.33) | 22.13 (+9.06) | 22.00 (+8.93) | 24.27 (+3.60) | 26.00 (+5.33) | 25.87 (+5.20) |
| | FDA-$L^0$ | 58.00 | 65.90 | 46.40 | 12.53 (+1.73) | 14.40 (+3.60) | 14.40 (+3.60) | 20.80 (+1.20) | 21.33 (+1.73) | 21.07 (+1.47) |
| | FDA-$L^{0-1}$ | 58.10 | 66.80 | 46.10 | 12.67 (+1.20) | 13.60 (+2.13) | 13.06 (+1.59) | 20.26 (-0.14) | 19.87 (-0.53) | 20.13 (-0.27) |
| | FDA-$L^{all}$ | 57.90 | 65.80 | 46.40 | 12.27 (+2.14) | 12.93 (+2.80) | 13.30 (+3.17) | 20.53 (-0.27) | 21.60 (+0.80) | 21.60 (+0.80) |
| 4/255 | No Defense | 58.50 | 65.90 | 46.30 | 17.47 (+7.07) | 20.40 (+10.00) | 20.40 (+10.00) | 24.40 (+3.47) | 26.53 (+5.60) | 27.30 (+6.37) |
| | TeCoA | 57.20 | 64.70 | 45.00 | 18.00 (+6.13) | 19.07 (+7.20) | 19.33 (+7.46) | 24.13 (+2.80) | 26.13 (+4.80) | 26.13 (+4.80) |
| | FARE | 56.40 | 64.20 | 44.70 | 18.93 (+5.86) | 21.47 (+8.40) | 22.00 (+8.93) | 24.27 (+3.60) | 26.27 (+5.60) | 25.87 (+5.20) |
| | FDA-$L^0$ | 58.00 | 65.90 | 46.40 | 13.07 (+2.27) | 14.40 (+3.60) | 14.40 (+3.60) | 20.53 (+0.93) | 20.53 (+0.93) | 20.80 (+1.20) |
| | FDA-$L^{0-1}$ | 58.10 | 66.80 | 46.10 | 12.80 (+1.33) | 13.33 (+1.86) | 13.33 (+1.86) | 20.67 (+0.27) | 19.87 (-0.53) | 20.13 (-0.27) |
| | FDA-$L^{all}$ | 57.90 | 65.80 | 46.40 | 12.27 (+2.14) | 13.20 (+3.07) | 13.47 (+2.67) | 20.13 (-0.67) | 21.87 (+1.07) | 21.73 (+0.93) |

## F  DETAILS FOR FUNCTION WORD DICTIONARY

We provide the function word dictionary we used as follows: "*am, is, are, was, were, be, been, being, have, has, had, do, does, did, will, would, shall, should, may, might, must, can, could, ought, dare, need, used, to, a, an, the, and, but, if, or, because, as, until, while, of, at, by, for, with, about, against, between, into, through, during, before, after, above, below, from, in, out, on, off, over, under, again, further, then, once, here, there, when, where, why, how, all, any, both, each, few, more, most, other, some, such, no, nor, not, only, own, same, so, than, too, very*".

## G  VISUALIZATION OF ATTENTION SCORES

Finally, we provide an illustration of original attention, FDA with one subtraction, and FDA, as shown in Fig.4.

Table 18: ASR of white-box *untargeted* attacks against **Text-to-Image Retrieval** on Flickr30k. The model is **ALBEF**. After-attack R@k values are presented in parentheses. All results are in percentage (%). ASR@1/5 indicates the drop of R@1/5 after attacks.

| $l_\infty$ | Defense | Clean ↑ | PGD | | APGD | | MAPGD | |
|---|---|---|---|---|---|---|---|---|
| | | | ASR@1 ↓ | ASR@5 ↓ | ASR@1 ↓ | ASR@5 ↓ | ASR@1 ↓ | ASR@5 ↓ |
| | No Defense | 95.90 | 78.54 (21.46) | 57.39 (42.61) | 74.70 (25.30) | 55.89 (44.11) | 70.93 (29.07) | 48.17 (51.83) |
| 2/255 | TeCoA | 91.20 | 81.22 (18.78) | 60.20 (39.80) | 76.02 (23.98) | 58.13 (41.87) | 67.95 (32.05) | 46.01 (53.99) |
| | TeCoA + FDA-$L^{0-1}$ | 91.60 | 80.73 (19.27) | 58.72 (41.28) | 68.87 (31.13) | 50.20 (49.80) | 67.75 (32.25) | 44.12 (55.88) |
| | FARE | 91.10 | 74.39 (25.61) | 51.52 (48.48) | 54.35 (45.65) | 70.95 (29.05) | 49.29 (50.71) | 23.79 (76.21) |
| | FARE + FDA-$L^{0-1}$ | 90.60 | 76.73 (23.27) | 53.46 (46.54) | 52.95 (47.05) | 69.31 (30.69) | 49.19 (50.81) | 26.32 (73.68) |
| | No Defense | 95.90 | 96.15 (3.85) | 92.00 (8.00) | 88.11 (11.89) | 76.73 (23.27) | 87.80 (12.20) | 72.97 (27.03) |
| 4/255 | TeCoA | 91.20 | 98.98 (1.02) | 95.33 (4.67) | 85.44 (14.56) | 73.91 (26.09) | 87.36 (12.64) | 71.49 (28.51) |
| | TeCoA + FDA-$L^{0-1}$ | 91.60 | 98.68 (1.32) | 94.73 (5.27) | 85.09 (14.91) | 72.41 (27.59) | 87.02 (12.98) | 69.47 (30.53) |
| | FARE | 91.10 | 98.08 (1.92) | 93.93 (6.07) | 80.57 (19.43) | 63.56 (36.44) | 80.57 (19.43) | 63.56 (36.44) |
| | FARE + FDA-$L^{0-1}$ | 90.60 | 98.17 (1.83) | 93.90 (6.10) | 79.67 (20.33) | 62.60 (37.40) | 80.18 (19.82) | 58.33 (41.67) |

Table 19: ASR of white-box *untargeted* attacks against **Image-to-Text Retrieval** on Flickr30k. The model is **ALBEF**. After-attack R@k values are presented in parentheses. All results are in percentage (%). ASR@1/5 indicates the drop of R@1/5 after attacks.

| $l_\infty$ | Defense | Clean ↑ | PGD | | APGD | | MAPGD | |
|---|---|---|---|---|---|---|---|---|
| | | | ASR@1 ↓ | ASR@5 ↓ | ASR@1 ↓ | ASR@5 ↓ | ASR@1 ↓ | ASR@5 ↓ |
| | No Defense | 85.60 | 84.85 (15.15) | 69.91 (30.09) | 77.34 (22.66) | 64.58 (35.42) | 76.02 (23.98) | 57.65 (42.35) |
| 2/255 | TeCoA | 81.44 | 87.68 (12.32) | 74.26 (25.74) | 74.70 (25.30) | 61.62 (38.38) | 74.49 (25.51) | 58.16 (41.84) |
| | TeCoA + FDA-$L^{0-1}$ | 81.80 | 88.59 (11.41) | 74.02 (25.98) | 73.91 (26.09) | 59.89 (40.11) | 73.26 (26.74) | 56.63 (43.37) |
| | FARE | 81.48 | 84.09 (15.91) | 69.48 (30.52) | 64.18 (35.82) | 74.06 (25.94) | 61.80 (38.20) | 41.13 (58.87) |
| | FARE + FDA-$L^{0-1}$ | 80.14 | 83.61 (16.39) | 68.98 (31.02) | 63.70 (36.30) | 73.27 (26.73) | 61.39 (38.61) | 61.25 (58.75) |
| | No Defense | 85.60 | 97.08 (2.92) | 93.61 (6.39) | 89.11 (10.89) | 81.30 (18.70) | 88.34 (11.66) | 77.89 (22.11) |
| 4/255 | TeCoA | 81.44 | 99.13 (0.87) | 96.51 (3.49) | 86.92 (13.08) | 79.35 (20.65) | 90.81 ( 9.19) | 81.08 (18.92) |
| | TeCoA + FDA-$L^{0-1}$ | 81.80 | 99.72 (0.76) | 97.61 (2.39) | 86.96 (13.04) | 78.91 (21.09) | 85.82 (14.18) | 80.33 (19.67) |
| | FARE | 81.48 | 98.27 (1.73) | 95.45 (4.55) | 84.96 (15.04) | 74.06 (25.94) | 84.96 (15.04) | 74.06 (25.94) |
| | FARE + FDA-$L^{0-1}$ | 80.14 | 98.35 (1.65) | 95.27 (4.73) | 83.83 (16.17) | 73.27 (26.73) | 84.93 (15.07) | 70.85 (29.15) |

Table 20: ASR of white-box *untargeted* attacks against **Text-to-Image Retrieval** on Flickr30k. The model is **BLIP**. After-attack R@k values are presented in parentheses. All results are in percentage (%). ASR@1/5 indicates the drop of R@1/5 after attacks.

| $l_\infty$ | Defense | Clean ↑ | PGD | | APGD | | MAPGD | |
|---|---|---|---|---|---|---|---|---|
| | | | ASR@1 ↓ | ASR@5 ↓ | ASR@1 ↓ | ASR@5 ↓ | ASR@1 ↓ | ASR@5 ↓ |
| | No Defense | 97.20 | 82.45(17.55) | 61.79 (38.21) | 80.94 (19.06) | 67.40 (32.60) | 72.22 (27.78) | 52.96 (47.04) |
| 2/255 | TeCoA | 81.50 | 55.33 (44.70) | 32.79 (67.21) | 47.73 (52.77) | 29.22 (70.78) | 44.16 (55.84) | 26.40 (73.60) |
| | TeCoA + FDA-$L^{0-1}$ | 80.40 | 51.30 (48.70) | 28.74 (71.26) | 44.25 (55.75) | 28.09 (71.91) | 40.56 (59.44) | 24.08 (75.92) |
| | FARE | 79.40 | 54.19 (45.81) | 28.29 (71.71) | 60.72 (+39.28) | 36.13 (63.87) | 58.65 (41.35) | 34.93 (65.07) |
| | FARE + FDA-$L^{0-1}$ | 79.30 | 51.41 (48.59) | 29.18 (70.82) | 56.62 (+43.38) | 33.08 (66.92) | 54.01 (45.99) | 30.59 (69.41) |
| | No Defense | 97.20 | 99.90 (0.10) | 99.60 ( 0.40) | 95.69 (4.31) | 90.57 (9.43) | 93.18 (6.82) | 83.75 (16.25) |
| 4/255 | TeCoA | 81.50 | 96.97 (3.03) | 93.94 ( 6.06) | 80.95 (19.05) | 68.40 (31.60) | 80.98 (19.02) | 66.38 (33.62) |
| | TeCoA + FDA-$L^{0-1}$ | 80.40 | 96.64 (3.36) | 92.73 ( 7.27) | 79.07 (20.93) | 65.08 (31.92) | 77.99 (22.01) | 63.81 (36.19) |
| | FARE | 79.40 | 94.23 (5.77) | 84.87 (15.13) | 85.09 (14.91) | 66.27 (33.73) | 83.46 (16.54) | 63.76 (36.42) |
| | FARE + FDA-$L^{0-1}$ | 79.30 | 94.14 (5.86) | 82.86 (17.14) | 82.43 (17.57) | 63.12 (36.88) | 80.04 (19.96) | 60.95 (39.05) |

Table 21: ASR of white-box *untargeted* attacks against **Image-to-Text Retrieval** on Flickr30k. The model is **BLIP**. After-attack R@k values are presented in parentheses. All results are in percentage (%). ASR@1/5 indicates the drop of R@1/5 after attacks.

| $l_\infty$ | Defense | Clean ↑ | PGD ASR@1 ↓ | PGD ASR@5 ↓ | APGD ASR@1 ↓ | APGD ASR@5 ↓ | MAPGD ASR@1 ↓ | MAPGD ASR@5 ↓ |
|---|---|---|---|---|---|---|---|---|
| | No Defense | 87.30 | 89.68 (10.32) | 78.74 (21.26) | 85.45 (14.55) | 74.51 (25.49) | 80.19 (19.81) | 65.22 (34.78) |
| 2/255 | TeCoA | 68.00 | 62.58 (37.42) | 41.35 (58.65) | 53.87 (46.13) | 34.11 (65.89) | 50.67 (49.33) | 31.41 (68.59) |
| | TeCoA + FDA-$L^{0-1}$ | 67.78 | 58.08 (41.92) | 37.06 (62.94) | 50.37 (49.63) | 30.10 (69.90) | 49.50 (50.50) | 28.36 (71.64) |
| | FARE | 65.64 | 63.35 (36.65) | 44.21 (55.79) | 68.64 (31.36) | 48.99 (51.01) | 66.25 (33.75) | 45.97 (54.03) |
| | FARE + FDA-$L^{0-1}$ | 66.22 | 59.73 (40.27) | 39.97 (60.03) | 64.06 (35.94) | 44.39 (55.61) | 61.54 (38.46) | 41.74 (58.26) |
| | No Defense | 84.02 | 99.90 (0.10) | 99.79 (0.21) | 96.70 ( 3.30) | 93.09 (6.91) | 94.01 (5.99) | 88.34 (11.66) |
| 4/255 | TeCoA | 68.00 | 97.42 (2.58) | 93.13 (6.87) | 82.94 (17.06) | 69.33 (30.67) | 80.98 (19.02) | 66.38 (33.62) |
| | TeCoA + FDA-$L^{0-1}$ | 67.78 | 96.64 (3.36) | 91.04 (8.96) | 80.22 (19.78) | 64.43 (35.57) | 77.99 (22.01) | 63.81 (36.19) |
| | FARE | 65.64 | 94.21 (5.79) | 88.16 (11.84) | 87.41 (12.59) | 76.07 (23.93) | 86.52 (13.48) | 74.31 (25.69) |
| | FARE + FDA-$L^{0-1}$ | 66.22 | 94.45 (5.55) | 88.40 (11.60) | 86.00 (14.00) | 72.89 (27.11) | 84.24 (15.76) | 71.37 (28.63) |

Table 22: Attack success rate (ASR) of *untargeted* PGD/APGD/MAPGD (masked APGD) against for **Visual Grounding** (VG) on RefCOCO+. After-attack accuracies are presented in parentheses. All results are in percentage (%). ASR indicates an accuracy drop after attacks.

| $l_\infty$ | Defense | Clean Performance Val_d | Clean Performance Test A | Clean Performance Test B | Test A Split (↓) PGD | Test A Split (↓) APGD | Test A Split (↓) MAPGD | Test B Split (↓) PGD | Test B Split (↓) APGD | Test B Split (↓) MAPGD |
|---|---|---|---|---|---|---|---|---|---|---|
| | No Defense | 58.50 | 65.90 | 46.30 | 17.40 (48.50) | 15.90 (50.00) | 15.30 (50.60) | 13.20 (33.10) | 7.40 (38.90) | 7.60 (38.70) |
| 2/255 | TeCoA | 57.20 | 64.70 | 45.00 | 8.20 (56.50) | 9.80 (54.90) | 9.80 (54.90) | 3.00 (42.00) | 4.60 (40.40) | 4.70 (40.30) |
| | TeCoA + FDA-$L^{all}$ | 57.10 | 64.90 | 45.30 | 8.30 (56.60) | 9.80 (55.10) | 9.70 (55.20) | 3.60 (41.70) | 4.80 (40.50) | 5.00 (40.30) |
| | FARE | 56.40 | 64.20 | 44.70 | 9.40 (54.80) | 11.80 (52.40) | 12.00 (52.20) | 2.90 (41.80) | 4.70 (40.00) | 4.60 (40.10) |
| | FARE + FDA-$L^{all}$ | 56.10 | 63.70 | 44.70 | 9.50 (54.50) | 10.90 (53.10) | 10.80 (53.20) | 3.40 (41.00) | 4.00 (40.40) | 4.10 (40.30) |
| | No Defense | 58.50 | 65.90 | 46.30 | 21.40 (44.50) | 18.70 (47.20) | 18.20 (47.70) | 14.90 (31.40) | 8.60 (37.70) | 8.80 (37.50) |
| 4/255 | TeCoA | 57.20 | 64.70 | 45.00 | 8.30 (56.40) | 12.50 (52.20) | 12.20 (52.50) | 2.90 (42.10) | 5.90 (39.10) | 6.30 (38.70) |
| | TeCoA + FDA-$L^{all}$ | 57.10 | 64.90 | 45.30 | 7.90 (57.00) | 11.30 (53.60) | 11.60 (53.30) | 3.40 (41.90) | 5.90 (39.40) | 6.10 (39.20) |
| | FARE | 56.40 | 64.20 | 44.70 | 9.40 (54.80) | 11.80 (52.40) | 13.60 (50.60) | 3.10 (41.60) | 5.60 (39.10) | 5.70 (39.00) |
| | FARE + FDA-$L^{all}$ | 56.10 | 63.70 | 44.70 | 9.50 (54.50) | 10.90 (53.10) | 12.20 (51.80) | 2.70 (41.70) | 5.10 (39.30) | 5.30 (39.10) |

Table 23: ASR of ablation studies against **Text-to-Image Retrieval** on Flickr30k. The model is **ALBEF**. Changes over unattacked values are presented in parentheses. All results are in percentage (%). ASR@1/5 indicates the attack success rate of the adversarial image showing up in the top-1/5 position of the targeted text queries.

| $l_\infty$ | Defense | Clean ↑ | PGD ASR@1 ↓ | PGD ASR@5 ↓ | APGD ASR@1 ↓ | APGD ASR@5 ↓ | MAPGD ASR@1 ↓ | MAPGD ASR@5 ↓ |
|---|---|---|---|---|---|---|---|---|
| | No Defense | 95.90 | 0.30 (+0.20) | 7.50 (+6.50) | 14.60 (+14.50) | 15.70 (+14.70) | 50.10 (+50.00) | 81.90 (+80.90) |
| | FDA - $\mathcal{T}$ | 95.10 | 0.40 (+0.40) | 6.40 (+6.20) | 20.70 (+20.70) | 23.70 (+23.50) | 4.80 (+ 4.80) | 28.30 (+28.10) |
| | FDA - $\mathcal{T}$ & $\mathcal{H}$ | 93.80 | 0.50 (+0.50) | 6.90 (+6.60) | 16.80 (+20.90) | 19.30 (+19.30) | 13.60 (+13.60) | 21.80 (+21.80) |
| | FDA - $\mathcal{H}$ | 95.60 | 0.10 (+0.10) | 7.30 (+6.60) | 12.10 (+12.10) | 13.40 (+12.70) | 43.60 (+43.60) | 73.90 (+73.20) |
| 2/255 | $L^{all}, H^{all}$ | 95.50 | 0.10 (+0.10) | 7.30 (+6.80) | 14.40 (+14.40) | 15.90 (+15.40) | 46.50 (+46.50) | 84.90 (+84.40) |
| | $L^{all}, H^{6-11}$ | 95.00 | 0.20 (+0.20) | 8.00 (+7.70) | 16.70 (+16.70) | 19.50 (+19.20) | 48.90 (+48.90) | 85.60 (+85.30) |
| | $L^{all}, H^{0-5}$ | 95.40 | 0.40 (+0.40) | 5.90 (+5.20) | 12.80 (+12.80) | 14.20 (+13.50) | 43.50 (+43.50) | 77.30 (+76.60) |
| | $L^{0}, H^{0-5}$ | 95.60 | 0.10 (+0.10) | 7.30 (+6.60) | 12.10 (+12.10) | 13.40 (+12.70) | 43.60 (+43.60) | 73.90 (+73.20) |
| | $L^{0-1}, H^{0-5}$ | 95.40 | 0.20 (+0.20) | 6.90 (+6.10) | 12.00 (+12.00) | 12.80 (+12.00) | 43.30 (+43.30) | 73.60 (+72.80) |
| | Full Dict | 95.40 | 0.40 (+0.40) | 5.90 (+5.20) | 12.80 (+12.80) | 14.20 (+13.50) | 43.60 (+43.60) | 77.50 (+77.00) |
| | Shortlisted Dict | 95.10 | 0.30 (+0.30) | 6.60 (+6.10) | 12.80 (+12.80) | 14.20 (+13.50) | 43.50 (+43.50) | 77.30 (+76.60) |
| | No Defense | 95.90 | 4.30 (+4.20) | 14.10 (+13.10) | 16.50 (+16.40) | 16.60 (+15.60) | 75.00 (+74.90) | 87.00 (+86.00) |
| | FDA - $\mathcal{T}$ | 95.10 | 5.10 (+0.20) | 20.00 (+19.80) | 24.80 (+24.80) | 25.50 (+25.30) | 12.10 (+12.10) | 40.00 (+39.80) |
| | FDA - $\mathcal{T}$ & $\mathcal{H}$ | 93.80 | 4.70 (+4.70) | 16.70 (+16.40) | 20.90 (+20.90) | 22.20 (+21.90) | 19.90 (+19.90) | 23.50 (+23.20) |
| | FDA - $\mathcal{H}$ | 95.60 | 2.90 (+2.90) | 13.50 (+12.80) | 13.90 (+13.90) | 14.10 (+13.40) | 69.00 (+69.00) | 82.40 (+81.70) |
| 4/255 | $L^{all}, H^{all}$ | 95.50 | 3.90 (+3.90) | 14.70 (+14.70) | 16.30 (+16.30) | 16.70 (+16.20) | 76.70 (+76.70) | 92.10 (+91.60) |
| | $L^{all}, H^{6-11}$ | 95.00 | 3.30 (+3.30) | 15.70 (+15.40) | 19.60 (+19.60) | 20.10 (+19.80) | 77.70 (+77.70) | 91.70 (+91.40) |
| | $L^{all}, H^{0-5}$ | 95.40 | 3.00 (+3.00) | 13.50 (+12.80) | 14.60 (+14.60) | 14.80 (+14.10) | 68.90 (+68.90) | 84.10 (+83.40) |
| | $L^{0}, H^{0-5}$ | 95.60 | 2.90 (+2.90) | 13.50 (+12.80) | 13.90 (+13.90) | 14.10 (+13.40) | 69.00 (+69.00) | 82.40 (+81.70) |
| | $L^{0-1}, H^{0-5}$ | 95.40 | 3.00 (+3.00) | 12.40 (+11.60) | 13.90 (+13.90) | 14.00 (+13.20) | 68.10 (+68.10) | 81.30 (+80.50) |
| | Full Dict | 95.40 | 3.00 (+3.00) | 13.50 (+12.80) | 14.90 (+14.90) | 15.20 (+14.70) | 69.60 (+69.60) | 83.90 (+83.40) |
| | Shortlisted Dict | 95.10 | 3.70 (+3.70) | 11.60 (+11.10) | 12.80 (+12.80) | 14.20 (+13.50) | 43.50 (+43.50) | 77.30 (+76.60) |

Table 24: ASR of ablation studies against **Image-to-Text Retrieval** on Flickr30k. The model is **ALBEF**. Changes over unattacked values are presented in parentheses. All results are in percentage (%). ASR@1/5 indicates the attack success rate of the adversarial image showing up in the top-1/5 position of the targeted text queries.

| $l_\infty$ | Defense | Clean↑ | PGD | | APGD | | MAPGD | |
|---|---|---|---|---|---|---|---|---|
| | | | ASR@1↓ | ASR@5↓ | ASR@1↓ | ASR@5↓ | ASR@1↓ | ASR@5↓ |
| | No Defense | 85.60 | 0.30 (+0.30) | 1.10 (+1.10) | 14.40 (+14.50) | 15.40 (+15.40) | 53.50 (+53.50) | 63.50 (+63.50) |
| | FDA - $\mathcal{T}$ | 85.28 | 0.30 (+0.30) | 1.50 (+1.50) | 20.90 (+20.90) | 22.30 (+22.30) | 5.90 (+ 5.90) | 10.40 (+10.40) |
| | FDA - $\mathcal{T}$ & $\mathcal{H}$ | 93.80 | 0.10 (+0.10) | 1.10 (+1.10) | 16.80 (+20.90) | 17.80 (+17.80) | 13.00 (+13.00) | 15.10 (+15.10) |
| | FDA - $\mathcal{H}$ | 85.50 | 0.10 (+0.10) | 0.60 (+0.60) | 12.30 (+12.30) | 12.80 (+12.80) | 46.50 (+46.50) | 56.20 (+56.20) |
| 2/255 | $L^{all}, H^{all}$ | 85.54 | 0.30 (+0.30) | 1.00 (+1.00) | 14.40 (+14.40) | 15.00 (+15.00) | 51.70 (+51.70) | 60.90 (+60.90) |
| | $L^{all}, H^{6-11}$ | 84.96 | 0.30 (+0.30) | 1.10 (+1.10) | 17.30 (+17.30) | 17.80 (+17.80) | 55.60 (+48.90) | 72.30 (+72.30) |
| | $L^{all}, H^{0-5}$ | 85.40 | 0.20 (+0.20) | 1.00 (+1.00) | 13.50 (+13.50) | 13.70 (+13.70) | 48.50 (+48.50) | 58.00 (+58.00) |
| | $L^0, H^{0-5}$ | 85.50 | 0.10 (+0.10) | 0.60 (+0.60) | 12.30 (+12.30) | 12.80 (+12.80) | 46.50 (+46.50) | 56.20 (+56.20) |
| | $L^{0-1}, H^{0-5}$ | 85.32 | 0.10 (+0.10) | 0.80 (+0.80) | 12.10 (+12.10) | 12.50 (+12.50) | 46.90 (+46.90) | 55.90 (+55.90) |
| | $L^{all}, H^{0-5}$ - Full Dict | 84.46 | 0.40 (+0.40) | 5.90 (+5.20) | 13.50 (+13.70) | 13.70 (+13.70) | 48.00 (+48.00) | 57.40 (+57.40) |
| | $L^{all}, H^{0-5}$ - Shortlisted Dict | 85.40 | 0.20 (+0.20) | 1.00 (+1.00) | 13.50 (+13.50) | 13.70 (+13.70) | 48.50 (+48.50) | 58.00 (+58.00) |
| | No Defense | 85.60 | 4.50 (+4.50) | 9.80 (+9.80) | 15.70 (+15.70) | 15.90 (+15.90) | 74.40 (+74.40) | 79.00 (+79.00) |
| | FDA - $\mathcal{T}$ | 95.10 | 6.40 (+6.40) | 11.30 (+11.30) | 24.50 (+24.50) | 24.90 (+24.90) | 15.50 (+15.50) | 19.10 (+19.10) |
| | FDA - $\mathcal{T}$ & $\mathcal{H}$ | 93.80 | 5.10 (+5.10) | 10.00 (+10.00) | 20.80 (+20.80) | 21.10 (+21.10) | 19.80 (+19.80) | 21.00 (+21.00) |
| | FDA - $\mathcal{H}$ | 85.50 | 3.30 (+3.30) | 6.50 (+ 6.50) | 13.70 (+13.70) | 13.70 (+13.70) | 68.80 (+68.80) | 72.40 (+72.40) |
| 4/255 | $L^{all}, H^{all}$ | 85.54 | 5.20 (+5.20) | 7.90 (+7.90) | 15.90 (+15.90) | 16.10 (+16.10) | 78.90 (+78.90) | 82.50 (+82.50) |
| | $L^{all}, H^{6-11}$ | 84.96 | 4.10 (+4.10) | 8.80 (+8.80) | 19.10 (+19.10) | 19.30 (+19.30) | 82.00 (+82.00) | 85.40 (+85.40) |
| | $L^{all}, H^{0-5}$ | 85.40 | 4.00 (+4.00) | 7.80 (+7.80) | 14.10 (+14.10) | 14.20 (+14.20) | 72.20 (+72.20) | 75.20 (+75.20) |
| | $L^0, H^{0-5}$ | 85.50 | 3.30 (+3.30) | 6.50 (+6.50) | 13.70 (+13.70) | 13.70 (+13.70) | 68.80 (+68.80) | 72.40 (+72.40) |
| | $L^{0-1}, H^{0-5}$ | 85.32 | 3.10 (+3.10) | 6.90 (+6.90) | 13.40 (+13.40) | 13.50 (+13.50) | 69.30 (+69.30) | 72.30 (+72.30) |
| | $L^{all}, H^{0-5}$ - Full Dict | 84.46 | 3.60 (+3.60) | 7.40 (+7.40) | 14.10 (+14.10) | 14.10 (+14.10) | 70.80 (+70.80) | 74.40 (+74.40) |
| | $L^{all}, H^{0-5}$ - Shortlisted Dict | 85.40 | 4.00 (+4.00) | 7.80 (+7.80) | 14.10 (+14.10) | 14.20 (+14.20) | 72.20 (+72.20) | 75.20 (+75.20) |

Table 25: Zero-shot performance by applying FDA as a plug-and-play tool on T2IR, I2TR on ALBEF/BLIP and VG on ALBEF.

| Tasks | Models | Method | Zero-shot Performance(↑) | | | | | | Average | |
|---|---|---|---|---|---|---|---|---|---|---|
| | | w/o FDA | 88.50 | 98.50 | 99.20 | 75.88 | 93.34 | 88.50 | 92.01 | - |
| | ALBEF | $L^{all}$ | 89.10 | 98.60 | 99.40 | 75.56 | 93.70 | 96.66 | 92.17 | ↑ 0.16 |
| | | $L^0$ | 89.00 | 98.50 | 99.30 | 75.38 | 93.20 | 96.72 | 92.02 | ↑ 0.01 |
| T2IR/I2TR | | $L^{0-1}$ | 89.60 | 98.80 | 99.40 | 76.16 | 93.70 | 96.80 | **92.41** | ↑ 0.40 |
| | | w/o FDA | 87.20 | 98.00 | 99.10 | 78.20 | 94.08 | 96.88 | 92.24 | - |
| | BLIP | $L^{all}$ | 88.70 | 98.40 | 99.30 | 78.80 | 94.20 | 96.88 | **92.71** | ↑ 0.47 |
| | | $L^0$ | 87.00 | 98.00 | 99.10 | 78.14 | 94.10 | 96.82 | 92.19 | ↓ 0.05 |
| | | $L^{0-1}$ | 87.10 | 98.00 | 99.10 | 78.12 | 94.16 | 96.82 | 92.22 | ↓ 0.02 |
| | | w/o FDA | 54.50 | | 61.77 | | 43.10 | | 53.12 | - |
| VG | ALBEF | $L^{all}$ | 54.73 | | 62.17 | | 43.11 | | **53.34** | ↑ 0.22 |
| | | $L^0$ | 54.10 | | 61.71 | | 42.34 | | 52.72 | ↓ 0.40 |
| | | $L^{0-1}$ | 54.14 | | 61.47 | | 42.42 | | 52.68 | ↓ 0.44 |

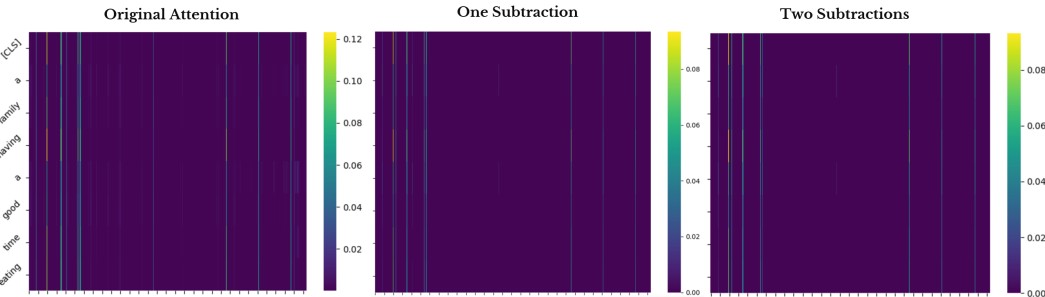

Figure 4: A heatmap of attention probabilities given the same image and text inputs. **Left**: Original attention probabilities are relatively 'noisy' and have several visible stripes with very low probabilities, implying the existence of some less relevant visual tokens that are activated, with negligible contributions. **Mid**: Attention probabilities with one FDA subtraction show much less aforementioned 'stripes', with much cleaner and more focused attentions. However, some distractions still exist and remain visible. **Right**: Attention probabilities with two subtractions show the cleanest attention maps and have the most negligible distractions, with only strong activations on the most relevant visual tokens, i.e., with higher probabilities.

