# OpenReview forum: "Pay Less Attention to Function Words for Free Robustness of Vision-Language Models"
_ICLR.cc/2026/Conference — ICLR 2026 Poster_

### Official Review · Reviewer_QdtA · 2025-10-23

**Soundness:** 2
**Presentation:** 2
**Contribution:** 3
**Rating:** 4
**Confidence:** 4

**Summary:**

This paper introduces Function-word De-Attention (FDA), a lightweight and novel method to improve the robustness of vision-language models (VLMs) without compromising their clean performance. The authors observe that function words (e.g., “the,” “is,” “and”) can make VLMs more vulnerable to cross-modal adversarial attacks because these words are frequent but semantically uninformative. FDA operates by computing and subtracting the cross-attention between function words and image tokens from the original attention maps. While the underlying mechanism behind FDA’s effectiveness is not yet fully understood, the authors provide extensive experiments demonstrating consistent improvements across multiple models, datasets, and attack settings.

**Strengths:**

1. Identifying function words as a source of vulnerability in VLMs is an original and well-motivated idea. Good observation.

2. FDA is conceptually straightforward and computationally efficient. It requires only a differential subtraction step within existing attention computations. It does not require additional parameters or retraining complexity.

3. The authors evaluate FDA across multiple models, datasets, and both targeted and untargeted attacks. The improvement over adversarial training baselines (TeCoA, FARE) is consistent and substantial.

**Weaknesses:**

1. It's an interesting observation that function words can lead as a source of vulnerability. However, the evidence provided are far from satisfying. Namely, a single sentence (“80.3% of images show higher similarity scores toward function words than content words after attacks”) and one visualization of distracted attention. It is insufficient to establish that this is a systematic rather than a coincidental effect. Additional qualitative and quantitative analyses would strengthen the claim. I also recommend including a baseline where function words are simply removed from the inputs, to better isolate and validate the proposed phenomenon.

2. Lack of clarity regarding experimental setup. FDA introduces a hyperparameter $\lambda$ to control the strength of “de-attention,” where $\lambda = 0$ reduces the method to standard attention. However, I find it confusing that the authors did not specify the $\lambda$ values used in the reported results. In particular, for the clean (non-attacked) setting, it is unclear whether the authors used a very small or even zero $\lambda$, which would make the claim of minimal performance drop less convincing. Moreover, the paper does not analyze how varying $\lambda$ affects performance and robustness, which is essential for understanding the method’s sensitivity and general applicability.

3. Several presentation issues. 1) Figure 1 seems to have noticeably lower resolution than other figures, even for texts in the figure. 2) Typo in Equation (4), $S_{T_f}$ instead of $S_{t_f}$. 3) Mistake in line 197-198, " Specifically, **TCL** shares the same backbones as **TCL** ..."

**Questions:**

I have made my suggestions along with weaknesses. Here are several open questions for the authors:

1. Have the authors examined whether the vulnerability arises specifically from function words, or more generally from irrelevant or low-information words in the inputs?

2. Could there be an adaptive mechanism to identify which words to down-weight or exclude, rather than relying on a fixed dictionary?

3. Are there cases where function words contribute meaningfully to visual grounding or alignment, and if so, how does FDA handle those instances?

---

> ### Author Response · Authors · 2025-11-21
> **Response for Official Review from Reviewer QdtA part 1/2.**
>
> **R1/Q1/Q2. Ablation\&validation of the vulnerability.**
> I.  Direct removal of function words.
> We understand your concern about whether the vulnerability comes from fiction words or simply low-information. To address your concern, we start by providing results on directly removing function words as comparison. We further apply masking on content words and nouns for sanity check. Each experiment follows the identical fine-tuning setting as FDA. We then test white-box robustness of PGD and APGD on all models. We did not conduct adaptive attacks for each control group due to their variation in defense mechanism.
>
> **Table.1**
> | Defense | |  Clean(R@1) ($\uparrow$) |    | ASR (2/255) ($\downarrow$)  | |   ASR (4/255) ($\downarrow$)  |  | Avg ASR drop $\Delta\_{ASR}$ ($\uparrow$) |
> | :---: | :---: | :---: | :---: | :---: | :---: | :---: | :---: | :---: |
> |  | **T2IR** | **I2TR** | **Avg** | **PGD** | **APGD** | **PGD** | **APGD** |  |
> | **w/o FDA** | 95.90 | 85.60 | 90.75 | 2.04 | 14.83 | 7.96 | 15.99 | \- |
> ||
> | **Cont Mask** | 21.50 | 11.10 | 16.30 | \- | \- | \- | \- | \- |
> | **Noun Mask** | 68.60 | 44.62 | 56.61 | \- | \- | \- | \- | \- |
> | **Func Mask** | 94.00 | 80.86 | 87.43 | 2.33 | 13.95 | 13.16 | 15.20 | $\uparrow$ 1.56 |
> ||
> | **FDA** | **95.60** | **85.50** | **90.55** | **1.86** | **12.50** | **6.40** | **13.70** | $\uparrow$ **23.07** |
>
> First of all, masking content words and nouns yields the largest performance drop, making it unviable for robustness evaluation. This aligns with the intuition that these words carry extensive semantic information crucial for VLM tasks. Furthermore, masking function words leads to evident performance drop (\~3%) and brings negligible robustness (\~1%), suggesting that simple removal of function words cannot weaken the reliance of VLMs on them. Nevertheless, FDA successfully removes these vulnerabilities and achieves the best clean performance and robustness, **confirming the superiority of attention subtraction over direct masking in enhancing robustness without causing performance drops**.
>
> II. Validation on removing low-similarity tokens.
> Furthermore, to specifically investigate the influence of low-information words, we use a cosine-similarity based mechanism (dubbed SIM) that selects the 2 words (tokens) with lowest image-text similarity (i.e., lowest semantic information regarding the image) prior to attention calculation, and apply the same attention subtraction as FDA accordingly for comparison. Results are given below.
>
> **Table.2**
> | Defense | | Clean R@1 ($\uparrow$)  |  |   $\Delta_{ASR}$ ($\uparrow$) |  Avg $\Delta_{ASR}$ ($\uparrow$)|
> | :---: | :---: | :---: | :---: | :---: | :---: |
> |  | T2IR | I2TR | $ 2/255$ | $5/255$ |  |
> | SIM | 95.20 | 85.32 | $\uparrow$ 9.54 | $\uparrow$ 8.02 | $\uparrow$ 8.78 |
> | FDA | **95.60** | **85.50** | $\uparrow$ **22.26** | $\uparrow$ **14.29** | $\uparrow$ **18.48** |
>
> Overall, the adaptive mechanism does not show advantages over both clean and adversarial examples, compared to using a fixed dictionary. Possible reasons could be that: *i)* the function word dictionary, brought from linguistic prior, could be more robust and approximate ground truth better for identifying vulnerability for subtraction; *ii)* the similarity-based method might potentially suffer from model bias and errors, causing instability for consistent vulnerability removal. In conclusion, the above results possibly implies that: while we cannot conclusively negate the correlation between the vulnerability and low semantic words, **function words exhibit stronger effectiveness in enhancing robustness than purely low-information words.**

---

> ### Author Response · Authors · 2025-11-21
> **Response for Official Review from Reviewer QdtA part 2/2.**
>
> **R2. Clarification on $\lambda$.**
> We apologize for the misunderstanding. $\lambda$ is **not an adjustable hyperparameter**. Instead, it is a learnable weight serving as a **control gate** to automatically control subtraction as part of our FDA module. All $\lambda$ is initialized to 1 as the fine-tuning begins, and we observe that it remains largely unchanged during and after training (stays between 0.98 and 1) for all models/layers. We have clarified this and renamed $\lambda$ as $\mathcal{G}$ to indicate its functionality as a control gate and modified Fig.2 accordingly in the revised pdf, which will be uploaded shortly.
>
> **R3. Typo.**
> We thank you for pointing out the typos and have corrected them accordingly. We will thoroughly proofread our paper for the revised version. Figure.1 is replaced with a clearer version, but the image resolution is fixed due to the dataset.
>
> **Q3. Meaningful function words.**
> We understand your concern when function words could occasionally carry certain semantic information. As presented in Table.5 and discussed in L409-416, we have compared the effectiveness of using the full dictionary (i.e., subtracting all function words) and using a shortlisted dictionary (i.e.,subtracting the “safest” ones). Results suggest that aggressively subtracting all function words also undermines the clean performance as well as the adversarial ones. Consequently, we use an expert shortlisted dictionary that only removes the safest function words, which can be found in Sec.E of the Appendix.
>
> We hope our responses address your concerns and are open to further discussion.

---

> ### Author Response · Authors · 2025-11-24
> **We loook forward to your responses.**
>
> Dear Reviewer QdtA,
>
> We have provided point-wise clarification and experiments (upon your request) for all your concerns. We look forward to your reply and hope that our responses could address your concerns. Your time and effert is very much appreciated.
>
> Regards,
>
> #1174 Paper Authors

---

> ### Comment · Reviewer_QdtA · 2025-11-25
> **Reply to Authors**
>
> Thank the authors for providing additional results and clarifications. I have carefully reviewed the authors’ response, the feedback from the other reviewers, as well as the updated manuscript. The paper’s main strengths lie in its insightful intuition, lightweight intervention, and comprehensive experimental evaluation.
>
> However, a major weakness, shared by all reviewers, remains the uncertain validity of the proposed heuristic. There is insufficient evidence that function words indeed contribute to vulnerability in vision-language models. For example, is it truly the case that all words in the expert-curated dictionary are detrimental, or could some play a useful role? Furthermore, it is unclear why such words would cause vision-language misalignment.
>
> The paper would be greatly strengthened by providing either a deeper theoretical or empirical explanation of how function words affect model robustness, or by developing a de-attention approach based on adaptively selected vocabularies (acknowledging the difficulty of this task). Additionally, I strongly recommend that the authors include the additional experiments in the manuscript. While these results are not entirely conclusive, they provide a useful teaser regarding the connection between function words and robustness.
>
> Overall, this is an interesting paper that demonstrates improved performance through thorough experimentation. I have updated my rating accordingly.

---

> > ### Author Response · Authors · 2025-11-26
> > **Clarification on our previous response.**
> >
> > Dear Reviewer QdtA,
> >
> > We sincerely appreciate your invaluable insights and feedback. Upon reading your reply, we identify a few unclarity in our previous response and would like to clarify accordingly:
> >
> > **... a de-attention approach based on adaptively selected vocabularies.**  We apologize for any misunderstanding and humbly emphasize that the part II. *Validation on removing low-similarity tokens* of our response for **R1/Q1/Q2** is a **sample-level adaptive de-attention** method upon your advice, i.e., **SIM** in the second table. Specifically, we utilize the cosine similarity of text/image embedding to adaptively identify the 2 tokens with the lowest similarity scores (excluding the [CLS] token). And the results show that FDA (with fixed dictionary) still surpasses SIM (with adaptive dictionary) regarding accuracy and robustness.
> >
> > **... include the additional experiments in the manuscript.** We appreciate your suggestions and assure all reviewers that **all ablation studies will be included jointly as a final revision** after carefully addressing all raised concerns before the end of rebuttal period.
> >
> > We are currently working on addressing the rest of your concerns and will provide corresponding responses shortly.
> >
> > #1174 Paper Authors

---

> ### Comment · Reviewer_QdtA · 2025-11-26
> **Thanks for further clarification**
>
> I am aware that applying text filtering based on similarity actually leads to worse performance. This is an interesting observation in itself because it suggests that similarity score is not a reliable metric for identifying vulnerability.
>
> What I mean in my previous response is that, if the authors were able to identify an appropriate metric for adaptively selecting text, and if this process ultimately required discarding function words proposed in the paper, then the overall contribution would be much easier for reviewers to accept. That said, I recognize that developing such a metric is likely nontrivial.
>
> Overall, I view this paper as an interesting first step toward that broader goal, and I appreciate the authors’ efforts in producing a clearer and more convincing revision.

---

> ### Author Response · Authors · 2025-12-03
> **Response for your latest concern- part 1/2.**
>
> (*All experiments are conducted on ALBEF on Flickr30k, retrieval.*)
>
> **R1. Variants of FDA.**
> We agree on adding extra variants of FDA as an orthogonal validation for the effectiveness of FDA. To choose the most representative ones from the many variants, we first record the proportion of all types of words, given as follows:
>
> **Table.1**
> | Word Type | NOUN | DET | VERB | ADJ | PRON | ADV | Others |
> | :---: | :---: | :---: | :---: | :---: | :---: | :---: | :---: |
> | Percentage| 32.10 | 18.24 | 11.93 | 9.23 | 2.34 | 1.12 | 25.04 |
>
> Hence, we choose determiners (DET) and adjectives (ADJ) regarding their proportions (NOUN\&VERB are not considered due to their semantic significance). DET indicates using a small subset (i.e., a/an/the) of function words, while ADJ adopts a completely different set of words. We follow the identical de-attentioning procedure for both variants, denoted as DDA and ADA, results are given as follows:
>
> **Table.2**
> | Defense | Clean |  |  |   Avg ASR Drop $\Delta_{ASR}$ $(\uparrow)$ ||
> | :---: | :---: | :---: | :---: | :---: | :---: |
> |  | T2IR | I2TR | $l_\infty=2/255$ | $l_\infty=4/255$ | Avg |
> | DDA | 95.60 | 85.42 | $\uparrow$14.08 | $\uparrow$13.96 | $\uparrow$ 9.28 |
> | ADA | 95.50 | 85.38 | $\uparrow$14.39 | $\uparrow$18.17 | $\uparrow$ 15.10 |
> | FDA | **95.60** | **85.50** | $\uparrow$ **27.61** | $\uparrow$ **18.53** | $\uparrow$ **23.07** |
>
> Overall, we find that **FDA leads the clean and adversarial performance** among other variants, i.e., DDA and ADA. Specifically, DDA, as a subset of FDA, shows almost identical clean performance, with a significant drop in robustness, indicating insufficient de-attentioning. ADA also shows subpar performance compared with FDA.
>
> **R2. Adaptive methods for word selection**.
> Finally, we further refine the adaptive mechanism based on text-image similarity (using per-token dot product of text and image features): instead of choosing fixed number of tokens, we adopt 3 implementations for adaptively selecting down-weighted tokens: i) setting threshold of $\mu - \delta$; ii) setting threshold of $\mu - 2\delta$, with $\mu,\delta$ being the mean and std of the text-image similarity. We further choose the lowest N tokens, with N being the number of function words in the texts. We denote them as SIM-$\delta$, SIM-2$\delta$, and SIM-N, respectively. Furthermore, we **record the % of selected words that are in our shortlisted function words dictionary**. Results are shown below.
>
> **Table.3**
> | Defense | % in the Dictionary | Clean |  | | ASR Drop $\Delta_{ASR} (\uparrow)$ | |
> | :---: | :---: | :---: | :---: | :---: | :---: | :---: |
> |  | | T2IR | I2TR | $l_\infty = 2/255$ |  $l_\infty = 4/255$ | Avg  |
> | SIM-N | 25.98%|95.90 | 85.50 |  $\uparrow$ 2.44 | $\uparrow$ 12.81  | $\uparrow$ 7.62 |
> | SIM-2$\delta$ | 74.53%|95.60 | 85.32 |  $\uparrow$ 9.38 | $\uparrow$ 13.86 | $\uparrow$ 8.39 |
> | SIM-$\delta$ | 79.47%|95.30 | 85.38 |  $\uparrow$ 12.85 | $\uparrow$ 11.54 | $\uparrow$ 12.41   |
> | FDA  |100%| 95.60 | 85.50 | $\uparrow$ **27.61** | $\uparrow$ **18.53** | $\uparrow$ **23.07** |
>
> In summary, we find that **the gained robustness of VLMs increased as the proportion of function words increased**. While we cannot design an adaptive mechanism that perfectly aligns with using the function words dictionary, we find that while the vulnerability of VLMs does not necessarily come from low-similarity (or low semantic) words, **there is an evident correlation between the percentage of function words and the gained robustness**.

---

> ### Author Response · Authors · 2025-12-03
> **Response for your latest concern- part 2/2.**
>
> **R3. Qualitative Validation**.
>
> To further validate the vulnerability brought by function words, we evaluate the performance drop/gain after removing nouns (**NOUN**), adjectives (**ADJ**), verbs (**VERB**) and function words (**FUNC**) regarding **clean** and **adversarial** examples. For adversarial examples, we use targeted APGD attacks for $\epsilon=2,4$. $\Delta_{ASR}$ is presented using the average results among T2IR and I2TR for all epsilons. Results are given below:
>
> **Table.4**
> | Masked Words | Clean |  | Avg ASR drop  $\Delta_{ASR} (\uparrow)$ |
> | :---: | :---: | :---: | :---: |
> |  | T2IR | I2TR |  |
> | **N/A** | 95.90 | 85.60 | \- |
> | **NOUN** | 58.90 | 32.83 | $\uparrow$ 25.27 |
> | **ADJ** | 91.60 | 78.36 | $\downarrow$ 0.42 |
> | **VERB** | 93.80 | 79.36 | $\downarrow$ 0.38 |
> | **FUNC** | 94.30 | 81.04 | $\uparrow$ 0.54 |
>
> First, removing NOUN lowers the ASR significantly (by over 25%), at the cost of greater performance drop on clean performances (over 30% and  50%, respectively). This aligns with intuition as NOUN carries most semantic information, and removing them destroys the semantic for both clean and adversarial examples. ADJ and VERB cause noticeable performance drops on clean and adversarial samples, implying that they make negligible contribution to semantic meaning for adversarial examples. Lastly, apart from NOUN, FUNC is the only word type that **lowers the ASR after removal**. Moreover, FUNC causes **the least drop on clean performance.** This confirms **the redundancy of function words regarding clean examples**, and validates its effectiveness in **incurring adversarial vulnerability**.

---

### Official Review · Reviewer_N7H5 · 2025-10-25

**Soundness:** 3
**Presentation:** 3
**Contribution:** 2
**Rating:** 4
**Confidence:** 4

**Summary:**

The paper introduces Function-word De-Attention (FDA) to improve the robustness and performance of VLMs by reducing attention to function words. FDA works by subtracting the cross-attention between function words and images from the original attention. Experimental results demonstrate that FDA reduces ASR with minimal or even improved performance.

**Strengths:**

1. The motivation and introduction of this paper is clear while use visualization to help readers quickly understand the task.

2. The experiments are thorough, and the performance is verified on three models, two tasks, and three datasets.

**Weaknesses:**

1. The authors' proposed operation, "proper removal of function words could potentially defend VLMs against such attacks," appears to have some limitations on VLM's use cases: the paper does not demonstrate its ability to defend against the fundamental task of vision-language models, such as classification. Furthermore, defense methods like FARE [1], which were originally designed to enhance the robustness of the LVLM's vision encoder, seems also inapplicable to this scenario.

2. In contribution summarization, the proposed “pioneer the theory” may be replaced with observation, motivation or other words. The paper does not use theory to prove that the difference in similarity after paying less attention can increase or decrease, or that the difference is bounded.

3. In Figure 2, how the conclusion “models can learn a more aligned cross-modal embedding” get? The observational evidence provided in the introduction doesn't suggest that this phenomenon is achieved by learning a more aligned cross-modal embedding. On the contrary, if the function word is removed, intuitively, the effect of alignment should be weakened.

4. Considering the trade-off between robustness and performance, it is recommended to compare with the TRADES [2] method.

[1] Schlarmann C, Singh N D, Croce F, et al. Robust clip: Unsupervised adversarial fine-tuning of vision embeddings for robust large vision-language models[J]. arXiv preprint arXiv:2402.12336, 2024.

[2] Zhang H, Yu Y, Jiao J, et al. Theoretically principled trade-off between robustness and accuracy[C]//International conference on machine learning. 2019: 7472-7482.

**Questions:**

See Weaknesses.

---

> ### Author Response · Authors · 2025-11-21
> **Response for Official Review from Reviewer N7H5 part 1/2.**
>
> **R1.1 Classification.**
> We understand your concerns of applying FDA on classification for fundamental evaluation of FDA. To the best of our knowledge, none of existing works have directly applied any of the included models, i.e., ALBEF, TCL and BLIP, for classification, including their original papers. These models require fine-tuning with a task-specific head, which is not directly applicable for classification as it is predominantly implemented on CLIP-like models in a zero-shot setting.
>
> Moreover,  we **have already evaluated FDA on both fundamental and advanced tasks**, i.e., T2IR/I2TR and VG. Specifically, T2IR/I2TR evaluates FDA performance on **fundamental** VLM tasks because it requires coarser alignment, while VG tests FDA performance on a more **advanced** task requiring finer-grain alignment.  Lastly, our experiment covers 3 models for 2 VLM tasks on 3 different datasets and demonstrates the generalizability of FDA across all settings.
>
> **R1.2 FARE inapplicable.**
> We respectfully disagree. Whatever encoders these methods are designed for, they a**ll focus on training an adversarially robust VLMs**. In other words, the different design focuses are simply the distinctiveness of different methods.  Besides, our FDA is **the first** and **the only** method designed by modifying cross-encoder during training, and FARE is one of the only similar, comparable and state-of-the-art methods. Lastly, all attacks are conducted through adversarially perturbed **images**, which directly targets **vision encoders**.
>
> **R2. “Theory” in contribution.**
> We will switch “we pioneer the theory” to “we identify that…”

---

> ### Author Response · Authors · 2025-11-21
> **Response for Official Review from Reviewer N7H5 part 2/2.**
>
> **R3. More aligned model for robustness.**
> We will further clarify that the statement in our paper *a more aligned cross-modal embedding* refers to a more robust alignment against adversarial attacks, i.e., **a tighter decision boundary**, as visualized in Fig.3. Furthermore, results on clean performances also suggest slight fluctuation across tasks, models and datasets, indicating minor influence of subtracting function-word attention.
>
> To further adress your concern, we’ve conducted extra ablation on directly masking function words. We further apply masking on content words and nouns for sanity check. Each experiment follows the identical fine-tuning setting as FDA. We then test white-box robustness of PGD and APGD on all models. We did not conduct adaptive attacks for each control group due to their variation in defense mechanism.
>
> **Table.1**
> | Defense | |  Clean(R@1) ($\uparrow$) |    | ASR (2/255) ($\downarrow$)  | |   ASR (4/255) ($\downarrow$)  |  | Avg ASR drop $\Delta\_{ASR}$ ($\uparrow$) |
> | :---: | :---: | :---: | :---: | :---: | :---: | :---: | :---: | :---: |
> |  | **T2IR** | **I2TR** | **Avg** | **PGD** | **APGD** | **PGD** | **APGD** |  |
> | **w/o FDA** | 95.90 | 85.60 | 90.75 | 2.04 | 14.83 | 7.96 | 15.99 | \- |
> ||
> | **Cont Mask** | 21.50 | 11.10 | 16.30 | \- | \- | \- | \- | \- |
> | **Noun Mask** | 68.60 | 44.62 | 56.61 | \- | \- | \- | \- | \- |
> | **Func Mask** | 94.00 | 80.86 | 87.43 | 2.33 | 13.95 | 13.16 | 15.20 | $\uparrow$ 1.56 |
> ||
> | **FDA** | **95.60** | **85.50** | **90.55** | **1.86** | **12.50** | **6.40** | **13.70** | $\uparrow$ **23.07** |
>
> First of all, masking content words and nouns yields the largest performance drop, making it unviable for robustness evaluation. This aligns with the intuition that these words carry extensive semantic information crucial for VLM tasks. Furthermore, masking function words leads to evident performance drop (\~3%) and brings negligible robustness (\~1%). suggesting that simple removal of function words cannot weaken the reliance of VLMs on them. Nevertheless, FDA successfully removes these vulnerabilities and achieves the best clean performance and robustness, **confirming the superiority of attention subtraction over direct masking in enhancing robustness without causing performance drops**.
>
> The above reullts aligns with your intuition as removing function words weakens alignment. Thus, FDA **does not remove** function words but only **subtract function-word attention** to make sure that VLMs pay less attention to them.
>
> **R4. TRADES as baseline.**
> As advised, we have applied TRADES for comparison, and we use identical settings as other baselines. Due to the limited time period, we only tested it on ALBEF for Flickr30k. Results are given below. **Our FDA remains SOTA over TRADES**. Specifically, TRADES causes great performance drops in both accuracy and robustness, indicating that TRADES is **inapplicable** for VLM robustness as it is designed for the pure vision domain.
>
> **Table.2**
> | Defense | | Clean R@1 ($\uparrow$)  |  |   $\Delta_{ASR}$ ($\uparrow$) |  Avg $\Delta_{ASR}$ ($\uparrow$)|
> | :---: | :---: | :---: | :---: | :---: | :---: |
> |  | T2IR | I2TR | $2/255$ | $4/255$ |  |
> | TRADES | 76.90 | 65.18 | $\downarrow$ 244.69 | $\downarrow$ 281.45 | $\downarrow$ 263.07 |
> | FDA | **95.60** | **85.50** | $\uparrow$ **22.26** | $\uparrow$ **14.29** | $\uparrow$ **18.48** |
>
> We hope our responses address your concerns and are open to any further dicussion.

---

> > ### Comment · Reviewer_N7H5 · 2025-11-28
> >
> > Thanks to the reviewers for their clarification, most of my concerns have been resolved. However, I still have some concerns about R1.2. I agree that the paper and the baseline methods aim to obtain an adversarially robust VLM. If the paper's baseline is [1] or [2] and is entirely concerned with tasks like T2IR/I2TR, then I consider this objective reasonable, as it defines its scope. But the authors chose TeCoA [3] and FARE [4]. As I understand it, after AT, they can still retain all the characteristics of VLM, such as the classification in [3] and the use of [3] and [4] as vision encoders for LVLM in [4]. However, compared to the baseline chosen by the authors, this approach seems limited, and I don't understand why made this choice.
> >
> > [1] Lu D, Wang Z, Wang T, et al. Set-level guidance attack: Boosting adversarial transferability of vision-language pre-training models[C]//Proceedings of the IEEE/CVF International Conference on Computer Vision. 2023: 102-111.
> >
> > [2] Gao S, Jia X, Ren X, et al. Boosting transferability in vision-language attacks via diversification along the intersection region of adversarial trajectory[C]//European Conference on Computer Vision. 2024: 442-460.
> >
> > [3] Mao C, Geng S, Yang J, et al. Understanding zero-shot adversarial robustness for large-scale models[J]. arXiv preprint arXiv:2212.07016, 2022.
> >
> > [4] Schlarmann C, Singh N D, Croce F, et al. Robust clip: Unsupervised adversarial fine-tuning of vision embeddings for robust large vision-language models[J]. arXiv preprint arXiv:2402.12336, 2024.

---

> ### Author Response · Authors · 2025-11-24
> **We loook forward to your responses.**
>
> Dear Reviewer N7H5,
>
> We have provided point-wise clarification and experiments (upon your request) for all your concerns. We look forward to your reply and hope that our responses could address your concerns. Your time and effert is very much appreciated.
>
> Regards,
> #1174 Paper Authors

---

> ### Author Response · Authors · 2025-12-03
> **Clarification on your latest confusion.**
>
> We need to emphasize that the references \[1,2\] provided are **attacks** against VLMs, not **defenses**. Furthermore, since our FDA focuses on multiple tasks including **T2IR**, **I2TR** and **Visual Grounding**, there are **no directly applicable methods** for comparison. To this end, we can only use **TeCoA** and **FARE** as they are the only two comparable SOTA methods for robust VLMs.

---

### Official Review · Reviewer_nHus · 2025-10-28

**Soundness:** 2
**Presentation:** 3
**Contribution:** 2
**Rating:** 2
**Confidence:** 4

**Summary:**

This paper hypothesizes that function words (e.g., am, is, are)—which are semantically vague and non-specific—make Vision-Language Models (VLMs) vulnerable to cross-modal adversarial attacks.
To address this, the authors propose Function-word De-Attention (FDA), which introduces a parallel attention path that computes the cross-attention between function words and images, and then subtracts this “distraction” from the original attention through a differential subtraction mechanism.

On retrieval tasks, FDA reduces the Attack Success Rate (ASR) by 18/13/53% on ALBEF/TCL/BLIP, with only 0.2–0.6% clean performance degradation.
On visual grounding, it achieves a 90% ASR reduction with a 0.3% accuracy improvement.
The robustness gain scales with model size (e.g., +54% ΔASR for BLIP) and even improves zero-shot performance by +0.4% without fine-tuning.

**Strengths:**

- Conceptually novel to focus on function words as a source of VLM vulnerability.
- Simple and efficient method. No additional parameters or adversarial training required.
- Consistent and strong improvement across multiple models, datasets, and tasks.

**Weaknesses:**

- The paper does not perform a true white-box attack against FDA.
While baselines are evaluated under full white-box settings (with gradient access), FDA’s differential subtraction module is hidden from the attacker.
The proposed Masked APGD (MAPGD) does not actually backpropagate through the FDA operation, so the comparison is not fair.

- The “observation” section is weak.
The reported 80.3% statistic lacks detail—what dataset, model, or attack setting?
Showing only one qualitative example in Fig. 1 is not convincing as empirical evidence.

- The paper does not include any ablation or control experiment on other word classes (e.g., content words, nouns, verbs, adjectives).

- Table 1 presentation is confusing: R@1 (↑, higher is better) and ASR (↓, lower is better) are shown in the same rows without clear separation.

- The comparison is too limited: only TeCoA/FARE trained with ε = 1/255 are considered. Robustness should be compared against baselines trained under the same threat levels (ε = 2/255 or 4/255), as standard adversarial training does.

**Questions:**

see weaknesses

---

> ### Author Response · Authors · 2025-11-21
> **Response for Official Review from Reviewer nHus.**
>
> **R1. Attacks not fully white-box.**
> We emphasize that **all** **attacks** (PGD, APGD, MAPGD) **are fully white-box**. Following common practices, attackers have complete access to all modules within the victim model since FDA is integrated in cross-attention modules of VLMs. Specifically, MAPGD is simply an APGD attack with all function words in the input texts masked. We will clarify this in the revised version.
>
> **R2. Observation.**
> Assuming you mean L37-40 of the introduction, the results are conducted on Flickr30k, performing PGD attacks with $\epsilon=4/255$ against text-to-image/image-to-text retrieval. These details will be updated into the revised pdf. Furthermore, the example in Fig.1 is **not for emprical evidence**, but an simple illustration to showcase how a successful defense works through removal of functionwords. Empirical evaluation of our FDA is conducted through **extensive experiments** on 3 datasets, 2 tasks and 3 models for both targeted and untargeted attacks.
>
> **R3. Ablation.**
> To address your concern, we’ve conducted extra ablation on directly masking function words. We further apply masking on content words and nouns for sanity check. Each experiment follows the identical fine-tuning setting as FDA. We then test white-box robustness of PGD and APGD on all models. We did not conduct adaptive attacks for each control group due to their variation in defense mechanism.
>
> **Table 1.**
> | Defense | |  Clean(R@1) ($\uparrow$) |    | ASR (2/255) ($\downarrow$)  | |   ASR (4/255) ($\downarrow$)  |  | Avg ASR drop $\Delta\_{ASR}$ ($\uparrow$) |
> | :---: | :---: | :---: | :---: | :---: | :---: | :---: | :---: | :---: |
> |  | **T2IR** | **I2TR** | **Avg** | **PGD** | **APGD** | **PGD** | **APGD** |  |
> | **w/o FDA** | 95.90 | 85.60 | 90.75 | 2.04 | 14.83 | 7.96 | 15.99 | \- |
> ||
> | **Cont Mask** | 21.50 | 11.10 | 16.30 | \- | \- | \- | \- | \- |
> | **Noun Mask** | 68.60 | 44.62 | 56.61 | \- | \- | \- | \- | \- |
> | **Func Mask** | 94.00 | 80.86 | 87.43 | 2.33 | 13.95 | 13.16 | 15.20 | $\uparrow$ 1.56 |
> ||
> | **FDA** | **95.60** | **85.50** | **90.55** | **1.86** | **12.50** | **6.40** | **13.70** | $\uparrow$ **23.07** |
>
> First of all, masking content words and nouns yields the largest performance drop, making it unviable for robustness evaluation. This aligns with the intuition that these words carry extensive semantic information crucial for VLM tasks. Furthermore, masking function words leads to evident performance drop (\~3%) and brings negligible robustness (\~1%). suggesting that simple removal of function words cannot weaken the reliance of VLMs on them. Nevertheless, FDA successfully removes these vulnerabilities and achieves the best clean performance and robustness, **confirming the superiority of attention subtraction over direct masking in enhancing robustness without causing performance drops**.
>
> **R4. Table presentation.**
> We have modified Table.1 for a clearer presentation.
>
> **R5. $\epsilon$ of TeCoA and FARE.**
> As we stated in L204-207, we use $\epsilon=1/255$ such that FARE and TeCoA could maintain close clean performance as FDA for a fairer robustness comparison. Training TeCoA and FARE with higher $\epsilon$ would undoubtedly increase robustness, but will also downgrade clean performance greatly, making comparison unfair. The attacking strength (2/255, 4/255) also directly follows existing settings as TeCoA and FARE for a fairer comparison.
>
> We hope our responses address your concerns and are open to any further dicussion.

---

> ### Author Response · Authors · 2025-11-24
> **We loook forward to your responses.**
>
> Dear Reviewer nHus,
>
> We have provided point-wise clarification and experiments (upon your request) for all your concerns. We look forward to your reply and hope that our responses could address your concerns. Your time and effert is very much appreciated.
>
> Regards,
> #1174 Paper Authors

---

> > ### Comment · Reviewer_nHus · 2025-11-24
> > **Reply to Authors**
> >
> > Thank you for taking the time and for your thorough response.
> >
> > **R1. Attacks not fully white-box.**
> >
> > I apologize for the confusion. Thank you for the clarification.
> >
> > **R2. Observation.**
> >
> > Yes, I meant L37-40 of the introduction, sorry for the ambiguity.
> > Nevertheless, I still find that the current evidence and explanation for why function words matter for robustness remain somewhat weak. Strengthening the theoretical justification or providing more direct empirical evidence in the main paper would, in my view, substantially improve the overall contribution.
> >
> > **R3. Ablation.**
> >
> > Thank you for sharing the additional results.
> > To better isolate the effect of function words, I would also expect a comparison between Function-word De-Attention (FDA) and other de-attention variants—for example, Adjective DA, Adverb DA, Preposition DA, Determiner DA, Conjunction DA, etc.
> > Since the current evidence for the special role of function words is not fully convincing to me, such analyses could provide clearer support.
> >
> > ---
> > Overall, several of my concerns have been alleviated, and I have accordingly adjusted my score.

---

> ### Author Response · Authors · 2025-12-03
> **Response for your latest concerns- part1/2.**
>
> (*All experiments are conducted on ALBEF on Flickr30k, retrieval.*)
>
> **R1. Variants of FDA.**
> We agree on adding extra variants of FDA as an orthogonal validation for the effectiveness of FDA. To choose the most representative ones from the many variants, we first record the proportion of all types of words, given as follows:
>
> **Table.1**
> | Word Type | NOUN | DET | VERB | ADJ | PRON | ADV | Others |
> | :---: | :---: | :---: | :---: | :---: | :---: | :---: | :---: |
> | Percentage| 32.10 | 18.24 | 11.93 | 9.23 | 2.34 | 1.12 | 25.04 |
>
> Hence, we choose determiners (DET) and adjectives (ADJ) regarding their proportions (NOUN\&VERB are not considered due to their semantic significance). DET indicates using a small subset (i.e., a/an/the) of function words, while ADJ adopts a completely different set of words. We follow the identical de-attentioning procedure for both variants, denoted as DDA and ADA, results are given as follows:
>
> **Table.2**
> | Defense | Clean |  |  |   Avg ASR Drop $\Delta_{ASR}$ $(\uparrow)$ ||
> | :---: | :---: | :---: | :---: | :---: | :---: |
> |  | T2IR | I2TR | $l_\infty=2/255$ | $l_\infty=4/255$ | Avg |
> | DDA | 95.60 | 85.42 | $\uparrow$14.08 | $\uparrow$13.96 | $\uparrow$ 9.28 |
> | ADA | 95.50 | 85.38 | $\uparrow$14.39 | $\uparrow$18.17 | $\uparrow$ 15.10 |
> | FDA | **95.60** | **85.50** | $\uparrow$ **27.61** | $\uparrow$ **18.53** | $\uparrow$ **23.07** |
>
> Overall, we find that **FDA leads the clean and adversarial performance** among other variants, i.e., DDA and ADA. Specifically, DDA, as a subset of FDA, shows almost identical clean performance, with a significant drop in robustness, indicating insufficient de-attentioning. ADA also shows subpar performance compared with FDA.
>
> **R2. Adaptive methods for word selection**.
> Finally, we further refine the adaptive mechanism based on text-image similarity (using per-token dot product of text and image features): instead of choosing fixed number of tokens, we adopt 3 implementations for adaptively selecting down-weighted tokens: i) setting threshold of $\mu - \delta$; ii) setting threshold of $\mu - 2\delta$, with $\mu,\delta$ being the mean and std of the text-image similarity. We further choose the lowest N tokens, with N being the number of function words in the texts. We denote them as SIM-$\delta$, SIM-2$\delta$, and SIM-N, respectively. Furthermore, we **record the % of selected words that are in our shortlisted function words dictionary**. Results are shown below.
>
> **Table.3**
> | Defense | % in the Dictionary | Clean |  | | ASR Drop $\Delta_{ASR} (\uparrow)$ | |
> | :---: | :---: | :---: | :---: | :---: | :---: | :---: |
> |  | | T2IR | I2TR | $l_\infty = 2/255$ |  $l_\infty = 4/255$ | Avg  |
> | SIM-N | 25.98%|95.90 | 85.50 |  $\uparrow$ 2.44 | $\uparrow$ 12.81  | $\uparrow$ 7.62 |
> | SIM-2$\delta$ | 74.53%|95.60 | 85.32 |  $\uparrow$ 9.38 | $\uparrow$ 13.86 | $\uparrow$ 8.39 |
> | SIM-$\delta$ | 79.47%|95.30 | 85.38 |  $\uparrow$ 12.85 | $\uparrow$ 11.54 | $\uparrow$ 12.41   |
> | FDA  |100%| 95.60 | 85.50 | $\uparrow$ **27.61** | $\uparrow$ **18.53** | $\uparrow$ **23.07** |
>
> In summary, we find that **the gained robustness of VLMs increased as the proportion of function words increased**. While we cannot design an adaptive mechanism that perfectly aligns with using the function words dictionary, we find that while the vulnerability of VLMs does not necessarily come from low-similarity (or low semantic) words, **there is an evident correlation between the percentage of function words and the gained robustness**.

---

> ### Author Response · Authors · 2025-12-03
> **Response for your latest concerns- part2/2.**
>
> **R3. Qualitative Validation**.
>
> To further validate the vulnerability brought by function words, we evaluate the performance drop/gain after removing nouns (**NOUN**), adjectives (**ADJ**), verbs (**VERB**) and function words (**FUNC**) regarding **clean** and **adversarial** examples. For adversarial examples, we use targeted APGD attacks for $\epsilon=2,4$. $\Delta_{ASR}$ is presented using the average results among T2IR and I2TR for all epsilons. Results are given below:
>
> **Table.4**
> | Masked Words | Clean |  | Avg ASR drop  $\Delta_{ASR} (\uparrow)$ |
> | :---: | :---: | :---: | :---: |
> |  | T2IR | I2TR |  |
> | **N/A** | 95.90 | 85.60 | \- |
> | **NOUN** | 58.90 | 32.83 | $\uparrow$ 25.27 |
> | **ADJ** | 91.60 | 78.36 | $\downarrow$ 0.42 |
> | **VERB** | 93.80 | 79.36 | $\downarrow$ 0.38 |
> | **FUNC** | 94.30 | 81.04 | $\uparrow$ 0.54 |
>
> First, removing NOUN lowers the ASR significantly (by over 25%), at the cost of greater performance drop on clean performances (over 30% and  50%, respectively). This aligns with intuition as NOUN carries most semantic information, and removing them destroys the semantic for both clean and adversarial examples. ADJ and VERB cause noticeable performance drops on clean and adversarial samples, implying that they make negligible contribution to semantic meaning for adversarial examples. Lastly, apart from NOUN, FUNC is the only word type that **lowers the ASR after removal**. Moreover, FUNC causes **the least drop on clean performance.** This confirms **the redundancy of function words regarding clean examples**, and validates its effectiveness in **incurring adversarial vulnerability**.

---

### Official Review · Reviewer_h995 · 2025-10-31

**Soundness:** 2
**Presentation:** 3
**Contribution:** 2
**Rating:** 2
**Confidence:** 4

**Summary:**

This paper proposes Function-word De-Attention (FDA), a training-free defense that subtracts attention from function words in VLMs (e.g., ALBEF, BLIP).
It shows robustness gains under image-space attacks without hurting clean performance. However, the idea is limited to cross-attention models, and lacks ablation evidence to support its claims.

**Strengths:**

1. The function-word “de-attention mechanism” is straightforward and easy to integrate into VLMs.
2. Zero training cost and minimal inference overhead.
3. Works stably across three representative fusion-based VLMs (ALBEF, BLIP, TCL).

**Weaknesses:**

1. Limited scope
    * FDA fundamentally requires cross-attention between text and image; thus, it cannot be applied to CLIP-style encoders or LLM-based multimodal models (LLaVA, InternVL).
    * Therefore, I think the paper’s title (“robustness of VLMs”) overstates generality.
2. Kind of trivial core observation
    * Figure 1’s finding — that attacks disrupt function-word attention but not content-word grounding — is unsurprising and follows directly from semantic anchoring.
3. Missing ablation
    * The simplest verification — masking function words entirely and measuring retrieval performance — is absent.
    * Without it, it’s unclear which is more important; “attention subtraction” or just “ignoring function words.”
4. Over-claiming “training-free”
    * Some sections mention optional fine-tuning for λ-scaling and per-layer adaptation, which contradicts the strict “zero-training” claim.

**Questions:**

Please refer to the Weaknesses section.

---

> ### Author Response · Authors · 2025-11-21
> **Last-minute and substantial review modifications.**
>
> Reviewer h995,
>
> We notice that you have **substantially modified your comments** at 23:27, 19th Nov (AoE). Besides, after reading your "reviesed" review, we also identified multiple comments that appear highly similar to those of other reviewers.
>
> To defend the code of conduct, we will **stick to the original review and ignore your current comments**, as we have spent a considerable amount of time on addressing conerns in your orignal comments.
>
> Lastly, we respectufully point out that **large, unexplained post-submission modifications undermine the scientific integrity and professional code of conduct in the ICLR review process**.
>
> #1174 Paper authors

---

> > ### Comment · Reviewer_h995 · 2025-11-21
> >
> > Dear Authors,
> >
> > I realized that I had misunderstood part of the paper in my original review, and I revised my comments to correct those issues. I apologize for making these changes without prior notice.
> >
> > During the revision process, I re-read the paper carefully and also reviewed the other reviewers’ comments. Although some of my points overlap with theirs, I did not copy their wording; I independently reached the same concerns. While revising I even considered increasing my score, but after inspecting Table 1 I remained unsure about the **fairness of the ε comparisons**, as Reviewer nHus also pointed out. If the table reflects an unfairly low adversarial budget for the baselines, it would be difficult for me to raise my rating without clarification.
> >
> > At this point, if the **ε values in the tables can be shown to reflect a fair comparison**, or if the **defense remains effective under a training-free setting** with an appropriately chosen λ, I am fully open to raising my score.
> >
> > Once again, I apologize for the late modifications, and I would appreciate it if you could take the time to respond thoroughly to these points.
> >
> > Best regards,
> > Reviewer h995

---

> ### Author Response · Authors · 2025-11-21
> **Response to your latest concern.**
>
> **1. Choice of $\epsilon$ for TeCoA and FARE**
>
> On the contrary, we use $\epsilon=1/255$ for a **fairer and more straightforward robustness comparison** as $\epsilon=1/255$ ensures that FARE and TeCoA maintain closer clean performance as FDA. Training TeCoA and FARE with higher $\epsilon$ would undoubtedly increase robustness, but will also downgrade clean performance greatly, making comparison **unfair and unviable**, i.e., a lower acc with higher robustness v.s. a higher acc with lower robustness. The attacking strength (2/255, 4/255) also directly follows existing settings for a fairer comparison.
>
>
>
> **2. Training-free robustness with proper $\lambda$**
>
> We need to clarify that:
>
> *a)* training-free robustness, while worth exploring,  **is not within any of our claims or contributions** as we clearly stated that our method requires finetuning (L223-224);
>
> *b)* We apologize for the misunderstanding. $\lambda$ is **not an adjustable hyperparameter**. Instead, it is a learnable weight serving as a **control gate** to automatically control subtraction as part of our FDA module. All $\lambda$ is initialized to 1 as the fine-tuning begins, and we observe that it remains largely unchanged during and after training (stays between 0.98 and 1) for all models/layers. We have clarified this and renamed $\lambda$ as $\mathcal{G}$ to indicate its functionality as a control gate and modified Fig.2 accordingly in the revised pdf, which will be uploaded shortly.
>
> Finally, other required ablation will be posted shortly in our complete responses. We hope our responses address your concerns and are open to any further dicussion.

---

> > ### Comment · Reviewer_h995 · 2025-11-21
> >
> > Thank you for the clarification.
> > I understand now that the λ values stay very close to 1 (around 0.98–1),  and your explanation about choosing ε to match clean performance is clear to me.
> > With these points clarified, I think my initial score (2) should be revisited,  and I’m open to raising it as I review the additional results.
> >
> > I’ll look over the remaining ablations, your responses to the other reviewers,  and the updated materials before making a final decision.
> > Thanks again for the detailed and helpful responses.

---

> ### Author Response · Authors · 2025-11-21
> **Follow-up responses for edited review.**
>
> **R1.Applicability.**
> We need to emphasize that our FDA is specifically designed for **fine-tuning VLMs** **with modified cross-attention** for free robustness, as we clearly stated in L475-477 of our paper. Our method remains applicable for a substantial part of VLM families, including Qwen, OpenFlamingo, etc. Lastly, the applicability of our FDA on CLIP-like models **does not alter or weaken our contribution and effectiveness on all included models, datasets and tasks**. Applying FDA on CLIP-like VLMs is a completely different category, which, although worth exploring, is considered **out of scope**.
>
> **R2/Q1.Ablation.**
> To address your concern, we’ve conducted extra ablation on directly masking function words. We further apply masking on content words and nouns for sanity check. Each experiment follows the identical fine-tuning setting as FDA. We then test white-box robustness of PGD and APGD on all models. We did not conduct adaptive attacks for each control group due to their variation in defense mechanism.
>
> **Table 1.**
> | Defense | |  Clean R@1 ($\uparrow$) |    | ASR (2/255) ($\downarrow$)  | |   ASR (4/255) ($\downarrow$)  |  | ASR drop $\Delta\_{ASR}$ ($\uparrow$) |
> | :---: | :---: | :---: | :---: | :---: | :---: | :---: | :---: | :---: |
> |  | **T2IR** | **I2TR** | **Avg** | **PGD** | **APGD** | **PGD** | **APGD** |  |
> | **w/o FDA** | 95.90 | 85.60 | 90.75 | 2.04 | 14.83 | 7.96 | 15.99 | \- |
> ||
> | **Cont Mask** | 21.50 | 11.10 | 16.30 | \- | \- | \- | \- | \- |
> | **Noun Mask** | 68.60 | 44.62 | 56.61 | \- | \- | \- | \- | \- |
> | **Func Mask** | 94.00 | 80.86 | 87.43 | 2.33 | 13.95 | 13.16 | 15.20 | $\uparrow$ 1.56 |
> ||
> | **FDA** | **95.60** | **85.50** | **90.55** | **1.86** | **12.50** | **6.40** | **13.70** | $\uparrow$ **23.07** |
>
> First of all, masking content words and nouns yields the largest performance drop, making it unviable for robustness evaluation. This aligns with the intuition that these words carry extensive semantic information crucial for VLM tasks. Furthermore, masking function words leads to evident performance drop (\~3%) and brings negligible robustness (\~1%). suggesting that simple removal of function words cannot weaken the reliance of VLMs on them. Nevertheless, FDA successfully removes these vulnerabilities and achieves the best clean performance and robustness, **confirming the superiority of attention subtraction over direct masking in enhancing robustness without causing performance drops**.
>
> **R3/R4/R5/Q2/Q3**. Addressed in our previous response.
>
> We hope our responses address all of your concerns.

---

> ### Author Response · Authors · 2025-11-24
> **We loook forward to your responses.**
>
> Dear  Reviewer h995,
>
> We have provided point-wise clarification and experiments (upon your request) for all your concerns. We look forward to your reply and hope that our responses could address your concerns. Your time and effert is very much appreciated.
>
> Regards,
>
> #1174 Paper Authors

---

### Author Response · Authors · 2025-11-21
**We thank all reviewers for their effort and advices.**

We appreciate all reviewers for their invaluable inputs and effortness in perfecting our paper. Overall, **all reviewers** recognize the **simplicity and novelty** of our FDA. Specifically, **reviewer** **QdtA** regards FDA as *an original and well-motivated idea*, while **reviewer nHus** thinks the idea *conceptually novel*. Besides, **reviewer h995, N7H5 and QdtA** regard FDA as a *simple, straightforward* and *efficient* method. Lastly, **all reviewers** recognize the **thoroughness of our experiments** and the **effectiveness of FDA** across models/datasets/tasks/attacks. Moreover, **Reviewer nHus** and **QdtA** agree that FDA has *consistent and strong improvement*. On the other hand, **reviewer h995, nHus,** and **QdtA** advise using direct removal of function words as comparison to further validate the effectiveness of FDA, which will be addressed as follows

---

### Author Response · Authors · 2025-11-22
**Revised pdf uploaded.**

Dear Reviewers,

We have uploaded a revised version of our paper. The modification is provided point-wise, as follows:

1. Added footnote for details of our statistical results. (Sec.1)

2. Paraphrased contribution 1 as requested by Reviewer N7H5. (Sec.1)

3. Reuploaded a clearer version of Fig.1 as requested by Reviewer QdtA.

4. Fixed typos in Equation(4). (Sec.3)

5. Clarified the functionality of $\lambda$ and renamed it to $\mathcal{G}$. (Sec.3, Eq.6)

6. Updated Fig.2 according to #5, and change *a more aligned cross-embedding* to *a more robust cross-embedding*, as requested by Reviewer N7H5.

7. Added clarification on white-box settings as requested by Reviewer nHus. (Sec.4 Attacks)

8. Updated Table 1 for clearer presentation as requested by Reviewer nHus.

9. Minor modification for better presentation. (Appendix)

10. Typo fixed in L803 and Table.23. (Appendix)

As for ablations, we appreciate the feedbacks from all reviewers and assure that **all ablation studies will be included jointly as a final revision** after carefully addressing all raised concerns before the end of rebuttal period.

Best,

#1174 Paper Authors

---

### Author Response · Authors · 2025-12-03
**Review&Discussion Summary- part 2/2.**

| Reviewer | Concerns and Corresponding Responses |  | Reviewer’s Reply |
| ----- | ----- | ----- | :---: |
| **N7H5** (most concerns **addressed**, but **cannot edit the score** due to the incident) | **Concern 1**: (W1 of the review) Should include fundamental VL tasks such as classification. **Response**:  (R1.1 of our rebuttal-part 1/2) ALBEF/TCL/BLIP have never been used for classification. IT2R/T2IR is a fundamental VL task, whereas VG is an advanced one. |  | Addressed. |
|  | **Concern 2:**  (W1 of the review) FARE is inapplicable for comparison as it focuses on vision encoder and FDA focuses cross encoder. **Response:** (R1.2 of our rebuttal-part 1/2) **FARE** and **TeCoA** are the **only** two SOTA methods. FDA/FARE/TeCoA are all  for robust VLMs through different approaches. |  | Asked for further clarification in **Concern 3\.** |
|  | **Concern 3:**  (latest comment of the reviewer) Confused why TeCoA and FARE are chosen as baselines. **Response:** (our latest rebuttal) The references \[1,2\] provided by the reviewer are **attacks** against **T2IR/I2TR,** while our FDA focuses on **defenses** for **Retrieval** and **Grounding. TeCoA** \[3\] and **FARE** \[4\] are the **only two SOTA** comparable **baselines** for robust VLMs. |  | Cannot reply. |
|  | **Concern 4:** (W2 of the review) Should replace “pioneer the theory".  **Response:** (R2 of our rebuttal-part 1/2) Changed to “we identify that …” in the revised pdf. |  | Addressed. |
|  | **Concern 5:** (W3 of th review) Needed clarification on why removing function words helps with better alignment. **Response:** (R3 of our rebuttal-part 2/2) Provided corresponding ablation studies by masking **3 different types** of words (content, noun and function) in Table.1. **FDA remains SOTA among all groups.** Clarified the *alignment* means *robust alignment*. |  | Addressed. |
|  | **Concern 6:** (W4 of the review) Should include TRADES as a new baseline. **Response:** (R4 of our rebuttal-part 2/2) Provided requested TRADES for comparison in Table.2, showing that TRADES perform significantly worse than FDA and is inapplicable for VLMs.  |  | Addressed. |
| **QdtA** (**Raised** from **4** to **6**) | **Concern 1**: (W1/Q1/Q2 of the review) Required ablations: **a\)** Directly removing function words; **b\)** An adaptive mechanism for selecting down-weighting tokens. **Response:** (R1/Q1/Q2 of our rebuttal-part 1/2) Provided ablation on masking different words in Table.1, and similarity-based adaptive de-attention in Table.2. **FDA retains its superiority in both experiments.** |  | Partially addressed, asked for further validation in **Concern 2**. |
|  | **Concern 2:**  (latest comment from the reviewer) Required further validation, e.g., using an appropriate metric for selecting tokens, which ideally selects function words. **Response:** (our latest response) ***a)*** We compared FDA with **2 different variants** of FDA (Adj DA, Det DA) in Table.2, and **FDA remains SOTA** among all groups.  ***b)*** We used **3 adaptive selection methods** and **recorded the percentage of function words** in Table.3. We find that **a larger proportion of function words yields better robustness.** ***c)*** We designed a qualitative validation by evaluating the influence of removing different words while inferencing adversarial examples, as shown in Table.4. Results show that **only removing function words while testing can qualitatively reduce ASR** without significant performance degradation. |  | Cannot reply. |
|  | **Concern 3:** (W2 of the review) Clarification on λ. **Response:** (R2 of our rebuttal \-part 2/2) λ is a control gate instead of a hyperparameter. |  | Addressed. |
|  | **Concern 4:** (W3of the review) Typos and Fig.1 image resolution. **Response:** (R3 of our rebuttal-part 2/2) All typo corrected in the revised pdf. |  | Addressed. |
|  | **Concern 5:** (Q3 of the review) How FDA handles exceptions when function words carry semantic meaning. **Response:** (Q3 of our rebuttal-part 2/2) We have considered these exceptions and used a shortlisted dictionary instead of full dictionary to avoid over-de-attentioning, as presented in Table.5, where a shortlisted dictionary performs better both on clean and adversarial examples. |  | Addressed. |

We once again appreciate your time and effort, and sincerely hope our summarization could help catching key points of all nested discussions.

---

### Author Response · Authors · 2025-12-03
**Review&Discussion Summary- part 1/2.**

We thank all reviewers and ACs for their invaluable service and inputs. Below is a point-wise tabularized summary for all reviews and discussion from all reviewers.

 **Review\&Discussion Summary**
**All reviewers** have actively engaged in the discussion and acknowledged **most of their concerns addressed**.

1. **Reviewer h995** acknowledged the original review had “***misunderstood** part of the paper*” and **edited the review** during the rebuttal. After clarification, **h995** agreed that “*initial score (2) should be **revisited**”* and was “*fully* *open to **raising** **my score”***.
2. **Reviewer nHus** stated that “*several of my concerns have been alleviated”* and **raised** the score from **2 to 4**.
3. **Reviewer N7H5** said that *“most of my concerns have been resolved*”, but **cannot edit the score** at the time of comment.
4. **Reviewer QdtA** appreciated our paper as “*an interesting first step toward that broader goal*” and **raised** the score from **4 to 6**.

We will provide a condensed version of each concern-response in the following table and hope the summarization could help capture the key points of all discussions. Note: *"Cannot reply" means that the reviewers cannot further comment after the incident.*


| Reviewer | Concerns and Corresponding Responses |  | Reviewer’s Reply |
| ----- | ----- | ----- | :---: |
| **h995**(**Edited** the original review and **agreed to raise** **score**)| **Concern 1:** (see comment by h995) The selection of $\epsilon=1/255$ for training TeCoA and FARE is not fair. **Response:**  (Point 1 our response titled *Response to your latest concern*) $\epsilon$ is selected to ensure closer clean performance for a **fairer** **robustness comparison**.   |  | Addressed. |
| | **Concern 2:**  (see comment by h995) Whether an appropriate  $\lambda$ could yield effective defense  under a training-free setting (`factual error`). **Response:** (Point 2 of our response titled *Response to your latest concern*) **Training-free robustness is not any of our claims**, and λ is a control gate instead of a hyperparameter. |  | Addressed. |
|  | **Concern 3:** (W1 of the original review) FDA cannot be applied to CLIP-style models. **Response:** (R1 of our response titled *Follow-up responses for edited review*) FDA is solely designed for cross-encoder and remains applicable for a wide range of VLMs. |  | Cannot reply. |
|  | **Concern 4:** (W2 of the original review) Asked for ablation studies of directly removing function words. **Response:** (R2/Q1 of our follow-up rebuttal) Provided corresponding ablation studies by masking **3 different types** of words (content, noun and function) in Table.1. **FDA remains SOTA among all groups.**  |  | Cannot reply. |
| **nHus** (**Raised** from **2** to **4**)  | **Concern 1**: (W1 of the review) Attacks (especially MAPGD) are not true white-box (`factual error`). **Response**: (R1 of our rebuttal) All attacks are fully white-box and we will further clarify in the paper. |  | Addressed. |
|  | **Concern 2:** (W2 of the review) More details on the statistical results in the introduction. **Response:** (R2 of our rebuttal) Results are collected on the retrieval dataset of Flickr30k, using PGD with $\epsilon=4/255$. |  | Addressed. |
|  | **Concern 3 :** (W3 of the review) Asked for ablation studies of directly removing function words. **Response:** (R3 of our rebuttal) Provided corresponding ablation studies by masking **3 different types** of words (content, noun and function) in Table.1. **FDA remains SOTA among all groups.**  |  | Addressed. |
|  | **Concern 4:** (latest comment from nHus) Asked for further ablation including more FDA variants to validate FDA, such as Determiner De-attention (DDA) and Adjective De-attention (ADA). **Response:** (our latest response) ***a)*** We compared FDA with **2 different variants** of FDA (Adj DA, Det DA) in Table.2, and **FDA remains SOTA** among all groups.  ***b)*** We used **3 adaptive selection methods** and **recorded the percentage of function words** in Table.3. We find that **a larger proportion of function words yields better robustness.** ***c)*** We designed a qualitative validation by evaluating the influence of removing different words while inferencing adversarial examples, as shown in Table.4. Results show that **only removing function words while testing can qualitatively reduce ASR** without significant performance degradation. |  | Cannot reply. |
|  | **Concern 5:** (W4 of the review) Table.1 in the paper needs clearer presentation. **Response:**  (R4 of our rebuttal) Modified Table.1 accordingly in the revised pdf. |  | Addressed. |
|  | **Concern 6:** (W5 of the review)  The selection of $\epsilon=1/255$ for training TeCoA and FARE is not fair. **Response:** (R5 of our rebuttal) $\epsilon$ is selected to ensure closer clean performance for a **fairer** **robustness comparison**.  |  | Addressed. |

---

### Author Response · Authors · 2025-12-03
**Final version of revised pdf has been updated.**

Dear Reviewers,

We have included **all** (except the results for TRADES) **ablation studies** and **qualitative evaluations** in the paper. Specifically, we incorporate the qualitative results in Section.1, the ablation on masking words and FDA variants in Section.4.3. And the rest results in Section.A of the Appendix.

Best,

#1174 Authors

---

### Meta-Review · Area_Chair_ckYh · 2026-01-08

**Summary:**

This paper proposes an interesting perspective on enhancing the adversarial robustness of vision–language models without causing noticeable performance loss. The key idea is to remove function-word attention in parallel at the cross-modal encoder, a method termed Function-word De-Attention (FDA). Experiments evaluate the FDA across three models and two tasks on three datasets, yielding good, consistent results. All reviewers recognized the novelty, admitted the simplicity of the proposed method, and further agreed on the consistency and effectiveness of the experimental results. The primary concerns shared among reviewers are more ablation studies, more supportive evidence for the motivation, and clarification on the technical details, which seem to have been well addressed by the authors’ point-to-point rebuttal results.

To be honest, this paper acquires very negative original scores. However, considering the comprehensive discussion (especially the clear attitude of reviewer nHus, h995, and QdtA) during the rebuttal period and the updated revision, I lean to accept this paper. Notably, I am recommending an acceptance but **I wouldn't mind if the paper gets rejected**.

**Reviewer Concerns:**

Basically, it seems most of the concerns are fully discussed and addressed. However, there are still some concerns from N7H5 are not fully address during discussion period.

**Reviewer Scores:**

I notice 3/4 reviewers (except N7H5) clearly express the intention of improve the rating.

---

### Decision · Program_Chairs · 2026-01-26

Accept (Poster)